# Fast-diffusing receptor collisions with slow-diffusing peptide ligand assemble the ternary parathyroid hormone–GPCR–arrestin complex

Jonathan Pacheco [1], Karina A. Peña[1], Sofya Savransky[1,2], Alexandre Gidon[1,6], Gerald R. V. Hammond [3], John Janetzko [4] & Jean-Pierre Vilardaga [1,2,5] ✉

The assembly of a peptide ligand, its receptor, and β-arrestin (βarr) into a ternary complex within the cell membrane is a crucial aspect of G protein-coupled receptor (GPCR) signaling. We explore this assembly by attaching fluorescent moieties to the parathyroid hormone (PTH) type 1 receptor (PTH$_1$R), using PTH as a prototypical peptide hormone, along with βarr and clathrin, and recording dual-color single-molecule imaging at the plasma membrane of live cells. Here we show that PTH$_1$R exhibits a near-Brownian diffusion, whereas unbound hormone displays limited mobility and slow lateral diffusion at the cell surface. The formation of the PTH–PTH$_1$R–βarr complex occurs in three sequential steps: (1) receptor and ligand collisions, (2) phosphoinositide (PIP$_3$)-dependent recruitment and conformational change of βarr molecules at the plasma membrane, and (3) collision of most βarr molecules with the ligand-bound receptor within clathrin clusters. Our results elucidate the non-random pathway by which PTH–PTH$_1$R–βarr complex is formed and unveil the critical role of PIP$_3$ in regulating GPCR signaling.

The PTH type 1 receptor (PTH$_1$R) is a class B GPCR mainly expressed in the cell membrane of bone and renal tubule cells, where it regulates mineral-ion (Ca$^{2+}$ and PO$_4^{3-}$) homeostasis in the body, as well as bone growth and repair in response to PTH and PTH-related protein (PTHrP)[1,2]. The PTH-bound PTH$_1$R activates heterotrimeric Gs and Gq proteins at the plasma membrane, inducing the production of cAMP (via Gαs), elevation of cytosolic Ca$^{2+}$ (via Gαq), increase of PtdIns(3,4,5)P$_3$, referred to as PIP$_3$, (via Gβγ liberated from activated Gq), and recruitment of β-arrestin 1 and 2 isoforms (βarrs) to form a ternary PTH–PTH$_1$R–βarr complex at the plasma membrane[3]. This ternary complex facilitates receptor internalization and subsequent endosomal cAMP signaling[4–6], which contributes to critical biological functions of the receptor, such as the elevation of serum Vitamin D (reviewed in ref. 2). We previously showed that the elevation of PIP$_3$ is crucial for the interaction between PTH$_1$R and βarrs in response to PTH[7]. However, how PIP$_3$ controls the coupling between PTH$_1$R and βarrs remains unknown. Here, we characterize the initial trajectories taken by individual peptide hormone, receptor, and arrestin molecules in the assembly of the ternary PTH–PTH$_1$R–βarr2 complex at the plasma membrane (PM). The βarr2 isoform was arbitrarily selected as PTH induces recruitment of βarr1 and βarr2 to PTH$_1$R with similar potencies (EC$_{50}$) and efficacies (maximal recruitment)[6,8,9]. This study

[1]Department of Pharmacology and Chemical Biology, University of Pittsburgh School of Medicine, Pittsburgh, PA 15261, USA. [2]Graduate Program in Molecular Pharmacology, University of Pittsburgh School of Medicine, Pittsburgh, PA 15261, USA. [3]Department of Cell Biology, University of Pittsburgh School of Medicine, Pittsburgh, PA 15261, USA. [4]Department of Molecular and Cellular Physiology, Stanford University School of Medicine, Stanford, CA 94305, USA. [5]U.S. Department of Veterans Affairs, Pittsburgh Healthcare System, Pittsburgh, PA 15240, USA. [6]Present address: Faculty of Mathematics and Natural Sciences, University of Oslo, Oslo, Norway. ✉e-mail: jpv@pitt.edu

supports a model where most of the PTH–PTH$_1$R–βarr2 complex assembly occurs within clathrin clusters and through a non-random mechanism dependent on the initial interaction of PIP$_3$ with βarr2.

## Results

### Single-molecule detection

To better determine how the PTH–PTH$_1$R–βarr2 complex assembles at the plasma membrane, we analyzed the trajectories and dynamics of PTH, PTH$_1$R, and βarr2 at the single-molecule level using single and dual color total internal reflection fluorescence (TIRF) microscopy in live cells. To this end, the N-terminal 34 amino acids fragment of PTH (PTH$_{1-34}$) was labeled in position 13 with tetramethylrhodamine (TMR) as previously characterized[10] and referred to as PTH$^{TMR}$ (Supplementary Table 1). PTH$_1$R was C-terminally fused to either monomeric NeonGreen (PTH$_1$R$^{mNG}$) or the infrared fluorescent protein, iRFP670 (PTH$_1$R$^{iRFP}$), and βarr2 was fused to mNG (βarr2$^{mNG}$) or the photo-activatable red monomeric mCherry1 protein (βarr2$^{PAmCherry}$). These modifications did not alter receptor signaling activity or βarr2$^{mNG}$ recruitment at the PM, as verified by recording typical time courses of cAMP production in response to PTH and βarr2$^{mNG}$ fluorescence intensity by TIRF following PTH stimulation (Supplementary Fig. 1). Using a low cell surface density of PTH$_1$R$^{mNG}$ (-1.3 molecules/μm$^2$/s) (Supplementary Fig. 2) and three criteria, we ensured that individual fluorescent spots imaged under TIRF illumination correspond to single molecules. The first was the sudden and irreversible decrease in spot fluorescence in one step indicative of a single molecule photobleaching (Supplementary Fig. 3A); the second was the diffraction-limited size of individual fluorescent spots centered around 200 nm as determined by the Gaussian fit, which is consistent with single-molecule imaging (Supplementary Fig. 3B, C); the third related to a single peak distribution of fluorescence intensity of the spots (Supplementary Fig. S3D)[11]. Under single molecule tracking conditions, we calculated the fluorophore lifetime in fixed cells to have an appropriate temporal timeframe for our observations. This ensures the dynamic behavior is captured with an optimal temporal resolution (Supplementary Fig. S3E).

### Intrinsic slow mobility of PTH

To determine the mobility of PTH$^{TMR}$, a series of TIRF experiments were conducted (Fig. 1A). We observed the attachment of PTH$^{TMR}$ to cells not expressing PTH$_1$R (mock cells), showing that the ligand could bind to the cell surface by itself (Fig. 1A, B, and Supplementary Fig. 4A). The TMR fluorophore did not bind to cells expressing the receptor, indicating that the PTH$^{TMR}$ binding to mock cells was an intrinsic property of the peptide ligand rather than the fluorophore (Fig. 1B and Supplementary Fig. 4B).

PTH$^{TMR}$ was also detected in areas not covered by cells, indicating non-specific PTH$^{TMR}$ binding to fibronectin-coated cover glass (Supplementary Fig. 4A, C, and Supplementary movie 1). To differentiate PTH$^{TMR}$ molecules bound to the cover glass from those in cells, we analyzed the diffusion properties of the ligand both inside and outside the cell (Fig. 1C and Supplementary Fig. 5A–C). The diffusion coefficient values of PTH$^{TMR}$ were similar in cells expressing PTH$_1$R cells and in mock cells ($D = 0.0018 \pm 0.0006$ and $0.0015 \pm 0.0001$ μm$^2$/s, respectively) but significantly ($P = 0.0008$) distinct from the value of PTH$^{TMR}$ non-specifically attached to the glass coverslip ($D = 0.00059 \pm 0.00011$ μm$^2$/s) and which are considered immobile (Fig. 1C). We thus used a threshold in our analysis to differentiate PTH$^{TMR}$ on coverslips from that on cells. We excluded all fluorescent spots with diffusion coefficient values below the threshold diffusion $D = 0.00059$ μm$^2$/s (horizontal dotted line in Fig. 1C) plus two standard deviations. The diffusion coefficient of all filtered PTH$^{TMR}$ trajectories was similar to the diffusion of trajectories that colocalized with PTH1R$^{mNG}$, confirming the accuracy of our analysis (Supplementary Fig. 5D, E).

To verify that the cell surface diffusion of PTH was independent of cell attachment to the coverslip, we measured PTH$^{TMR}$ diffusion in the top part of the cell (not attached to the glass coverslip). To this end, we employed a live-cell technique, initially developed to image apical membrane proteins of epithelial cells in combination with TIRF microscopy, which provides the best signal-to-noise ratio suitable for single-molecule imaging[12]. To conduct this experiment, the cells were cultured on coverslips, and PTH$^{TMR}$ was added and washed out to prevent leftover ligands from floating around. Then, the coverslip was held in a screw-down piston device to allow the adjustment of the top cell surface against the coverslip of a glass-bottom dish (Fig. 2A, B). We obtained comparable diffusion coefficient values of PTH$^{TMR}$ on both the bottom and upper parts of the cell membrane, thus validating our results obtained from the bottom side of cells. (Fig. 2C).

The slow diffusion properties of PTH$^{TMR}$ obtained in HEK293 cells expressing the recombinant PTH$_1$R were verified in human primary renal proximal tubule epithelial cells (RPTEC) and murine osteoblast precursor cells (MC3T3) expressing the native receptor. No significant differences in diffusion were observed between these different cell lines, thereby corroborating the consistent slow diffusion of PTH$^{TMR}$ (Fig. 2D). Using fluorescein as an alternative dye to label PTH (PTH$^{Fluo}$) did not change the diffusion properties of PTH when tested in HEK293 cells expressing PTH$_1$R$^{iRFP}$ (Fig. 2E). Like PTH$^{TMR}$, PTH$^{Fluo}$ decreased the diffusion of PTH$_1$R when tested in cells expressing PTH$_1$R$^{iRFP}$ (Figs. 2F, 3F).

Next, we verified the motion of PTH$^{TMR}$ at the ensemble level by using fluorescence recovery after photobleaching (FRAP) in both

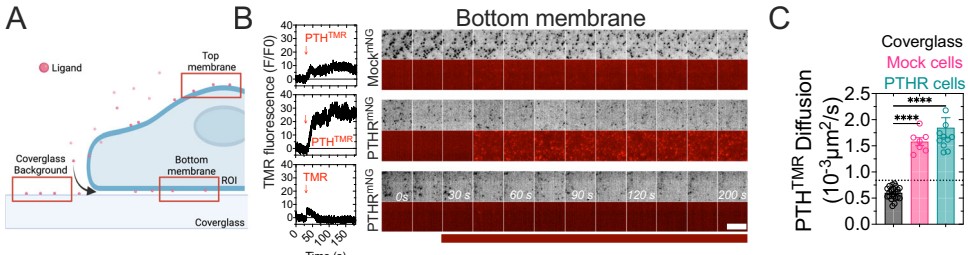

**Fig. 1 | Plasma membrane binding of PTH$^{TMR}$. A** Principle of TIRF experiments. The measurements were performed in 3 different regions of interest (ROI); the bottom membrane, the top membrane, and the cover glass. Created in BioRender. Pacheco Romero, J. (2025) https://BioRender.com/u84v021. **B** Fluorescence time courses (left) and representative images (right) of CLTA$^{mNG}$ cells without the expression of the receptor (mock cells) or cells expressing the PTH$_1$R$^{mNG}$. PTH$^{TMR}$ or TMR fluorophore was added at 30 sec, indicated by the horizontal red bar at the bottom. The mNG channel (in gray) was used to ensure the TIRF focus plane. Experiments are

representative of $n = 6/3$ (PTHR), $n = 5/4$ (mock) and $n = 5/2$ cells/independent days of experiment. Scale bar, 5 μm. **C** Diffusion coefficient comparison between PTH$^{TMR}$ molecules detected on cover glass with no cells, and HEK293 cells with or without expression of PTH$_1$R$^{mNG}$. The dotted line represents the threshold under which trajectories are excluded from our analysis. Grand mean ± 95% confidence intervals (C.I.) of $n = 10/179$ (PTH$_1$R$^{NG}$), $n = 23/829$ (cover glass), and $n = 7/3631$ (mock) with cells/average trajectories per cell. ****$P < 0.0001$ by one-way ANOVA with a Tukey's multiple comparison.

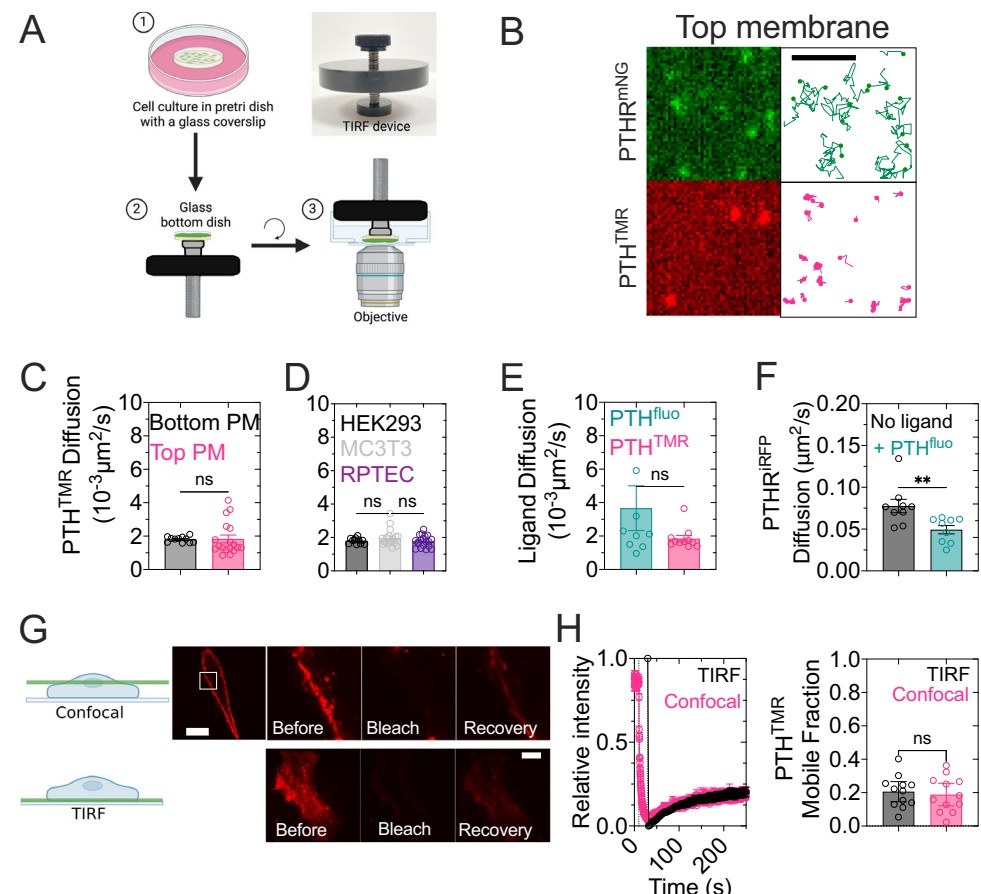

**Fig. 2 | Single molecule measurements of PTH^TMR. A, B** 3D-printed device for imaging the top cell surface by TIRF. The black lid with a screw-type plunger fits in the glass-bottom dishes. The plunger holds the sample coverslip down and ensures its correct adjustment at the bottom of the dish while preventing compression of cells (**A**). Representative trajectories of receptor and ligand at the top membrane (**B**). Scale bar, 2 µm. Scheme was created in BioRender. Pacheco Romero, J. (2025) https://BioRender.com/p78d742. **C** Comparison of diffusion of PTH^TMR in the bottom and top membranes. Data are the mean ± S.E.M. of Bottom plasma membrane (PM) (black), $n = 11/5751$, and top PM (teal), $n = 18/156$ cells/average trajectories per cell. ns, not significant with $P = 0.92$ by unpaired $t$-test. **D** Comparison of diffusion of PTH^TMR in HEK293 cells expressing PTH₁R^mNG (black), for comparison, the same data as panel C is shown (bottom PM), MC3T3 (gray), $n = 18/686$ and RPTEC cells, $n = 19/604$ cells/average trajectories per cell. Data are the mean ± S.E.M. ns, not significant by one-way ANOVA with a Tukey's multiple comparison. **E** Diffusion

coefficient of trajectories of PTH labeled with either TMR or fluorescein in cells expressing PTH₁R^mNG or PTH₁R^iRFP, respectively. Only trajectories that showed coincidental detection with the receptor were considered for the analysis. Mean ± S.E.M. of $n = 9/75$ (PTH^fluo) and $n = 12/777$(PTH^TMR) cells/average trajectories per cell. ns, not significant by unpaired $t$-test. **F** Diffusion coefficient of PTH₁R^iRFP in presence or absence of PTH^fluo. Mean ± S.E.M. of $n = 9/3674$ cells/average trajectories per cell. **P = 0.0086 by unpaired $t$-test. **G, H** FRAP recordings in confocal and TIRF microscopy for PTH^TMR. A representative cell before, after, and recovering from photobleaching. The scheme depicts the optical section that was selected for photobleaching. Scale bar, 10 µm. Time courses of fluorescence recovery and quantification of mobile fraction from FRAP experiments (**H**). Mean ± 95% C.I. of $n = 12$ cells of $N = 3$ experiments. ns, not significant by unpaired $t$-test. Scheme created in BioRender. Pacheco Romero, J. (2025) https://BioRender.com/i51c035.

confocal and TIRF microscopy. In confocal experiments, we selected a portion of the equatorial part of cells that do not include the basal membrane in contact with the coverslip (Fig. 2G). Only a modest recovery of fluorescence was detected for PTH^TMR (≈20%), indicating that a large proportion of ligands was not freely diffusing (Fig. 2H). These results were confirmed by using FRAP in the TIRF configuration. In this setup, the laser beam was focused on the cell membrane to induce the photobleaching of fluorophores on the basal membrane adjacent to the coverslip. The fluorescence recovery was monitored over time, corroborating a mobile fraction of 20% (Fig. 2H). These results indicate that slow cell surface diffusivity is an intrinsic property of PTH.

To delineate the part of PTH that contributes to its attachment to the cell surface, we recorded and compared the binding of TMR-labeled PTH, LAPTH, M-PTH₁₋₁₄, and the inverse agonist PTHrP^TMR_{7−36} (IA-PTH)[13] in single mock HEK293 cells (Supplementary Fig. 6A). To this end, ligand dissociation was determined through temporal changes of PTH^TMR emission upon rapid perfusion and washout of 1 µM ligand[10].

M-PTH₁₋₁₄^TMR exhibited the weakest attachment to the cell surface, given that <3% of the perfused ligand remained bound as compared to ≈25% for LAPTH^TMR and IA-PTH^TMR and 15% for PTH (Supplementary Fig. 6A). These results suggest that the carboxy-terminal portion of PTH is a determinant for cell surface binding. This part of PTH contains a potential BXXXXXB motif (where B is any basic residue and X is any hydrophobic residue) encountered in proteins binding glycosaminoglycans (GAGs) such as the hyaluronic acid (HA)[14], which is an essential component of the extracellular matrix (ECM)[15,16]. To test the role of HA in PTH^TMR binding to cells, we treated mock cells with hyaluronidase (HAase) to remove HA from the cell surface and measure time-resolved changes in TMR fluorescence in single cells using a photometric method. Our findings reveal that mock cells treated with HAase exhibited significantly less bound PTH^TMR when compared to untreated cells (Supplementary Fig. 6B). Additionally, a significant reduction in the density of single PTH^TMR molecules was observed at the surface of HAase-treated cells (Supplementary Fig. 6C). Although these results suggest that HA contributes to the retention of PTH at the

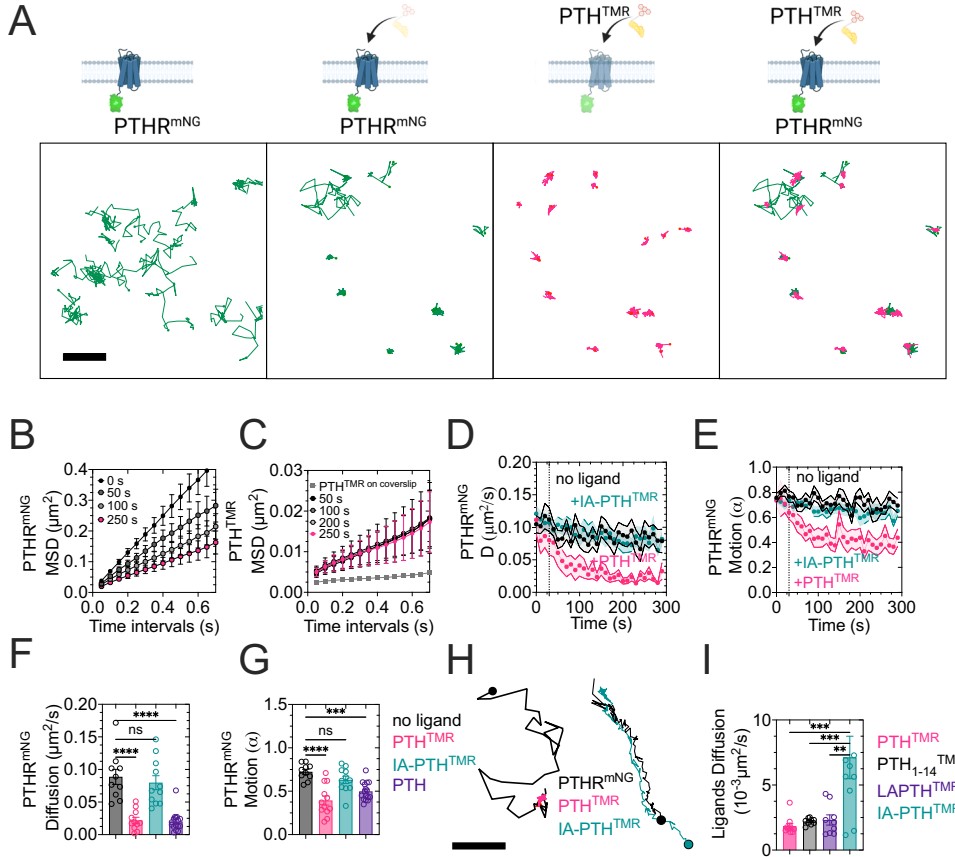

**Fig. 3 | Diffusion modes of PTH and PTH₁R. A** Scheme illustrating both labeled PTH₁R and peptide ligand used to record dual-color single molecule imaging (upper), and representative trajectories of PTH₁RmNG before (green-left) and after (green-center) stimulation with PTHTMR (magenta-center). The merge of trajectories depicts the coincidental detection of PTH₁RmNG and PTHTMR (right). Scale bar: 2 μm. Scheme created in BioRender. Pacheco Romero, J. (2025) https://BioRender.com/q57m113. **B**, **C** MSD *vs* time intervals plots of PTH₁RmNG (**B**) and PTHTMR (**C**). Time points indicate the cohorts of trajectories after adding PTHTMR over which mean squared displacement was calculated for each particle. Data are mean ± S.E.M. of *n* = 12 cells for both, PTH₁RmNG and PTHTMR. **D**, **E** Diffusion (**D**) and motion (**E**) time-lapses of PTH₁RmNG alone (black), in the presence of PTHTMR (magenta) or the inverse agonist, IA-PTHTMR (teal). Individual trajectories were analyzed at intervals of 10 s. Ligands were added at *T* = 30 s (vertical dotted line). Data are the mean ± S.E.M. of *n* = 11 cells derived from the same experiment used to calculate panels (**B**, **C**).

**F**, **G** Averaged diffusion coefficient (**F**) and motion (**G**) of PTH₁RmNG before and after PTHTMR, inverse agonist, or unlabeled PTH. Data are the mean ± S.E.M. of *n* = 10/1081 (PTH₁RmNG), *n* = 12/1481 (PTH₁RmNG + PTHTMR), *n* = 11/3027 (PTH₁RmNG + IA-PTHTMR) and *n* = 18/143 (PTH₁RmNG + PTH), cells/average trajectories per cell of *N* = 3 independent experiments. ****$P < 0.0001$ and ns, not significant ($P = 0.8633$ in F and $P = 0.39$ in G) by one-way ANOVA with a Tukey's multiple comparison. **H** Example of interacting PTH₁RmNG–PTHTMR or PTH₁RmNG– IA-PTHTMR trajectories. Scale bar: 0.5 μM. **I** Diffusion coefficients of agonists PTHTMR (*n* = 12/777), PTH₁₋₁₄TMR (*n* = 11/1562), LAPTHTMR (*n* = 9/824), and the inverse agonist IA-PTHTMR (*n* = 11/619). Only the trajectories that showed coincidental detection with PTH₁RmNG were considered for this analysis. Data are mean ± S.E.M. *n* = cells/average trajectories per cell.
****$P < 0.0001$, ***$P < 0.001$, **$P < 0.01$ by one-way ANOVA with a Dunnett multiple comparison.

cell surface, the truncated peptide (IA-PTH) experiment may not be the perfect test of the hypothesis regarding immobilization mediated by the BXXXXXXB motif; further testing will be required in the future.

## Lateral PTH₁R diffusion drives the assembly of the PTH−bound receptor complex

Compared to the slow diffusion of PTH, the PTH₁R exhibited dynamic lateral movement before adding PTHTMR. Following stimulation with PTHTMR, a drastic reduction in receptor mobility was observed (Fig. 3A). The displacement of individual PTH₁R molecules was linearly proportional to the mean-square displacement (MSD) over time (Fig. 3B). This process was well described by the equation MSD = 4Dtᵅ, where D is the diffusion coefficient. The power-law exponent α measures the linearity representing the type of motion with α ∼ 1 for Brownian (i.e., free) motion, α < 1 for a sub-diffusive (i.e., confined) motion, and α > 1 for a super-diffusive (i.e., directed) motion. To determine the effect of PTH on PTH₁RmNG mobility, we measured the diffusion coefficient of individual trajectories and calculated the average diffusion coefficient of

all trajectories registered in 10 s intervals over 5 min recordings. The MSD plot shows a reduction in PTH₁RmNG mobility after PTHTMR exposure (Fig. 3B). In contrast, the mobility of PTHTMR remained the same over the 5 min recordings and was distinct from the background ligand attached to the coverslip in absence of cells (Fig. 3C). The average diffusion coefficient (D) of PTH₁RmNG molecules was 0.088 μm²/s with a moderate sub-diffusive motion (α = 0.72 ± 0.03) in the absence of the ligand (Fig. 3D–G). The presence of PTHTMR significantly shifted the lateral movement of PTH₁RmNG molecules toward a reduced diffusion with D = 0.020 ± 0.004 μm²/s and increased sub-diffusivity with α = 0.33 ± 0.05 (Fig. 3D–G). Labeling the ligand did not affect PTH₁R diffusion, given that unlabeled PTH decreased diffusion and motion of the receptor as PTHTMR (Fig. 3F, G). FRAP confirmed the reduction in the diffusion of PTH₁RmNG after PTH treatment at the ensemble level. A faster fluorescence recovery was recorded in the absence of PTH than after its addition (Supplementary Fig. 7A). Notably, the mobile fraction of the receptor in the presence of PTH (Supplementary Fig. 7B) was like the mobile fraction of the peptide (∼20%) (Fig. 2H).

It is important to emphasize that the FRAP results indicate a phenomenon different from the single-molecule (SM) data. In FRAP experiments, the receptor is overexpressed to ensure there is enough ligand at the cell surface. On the other hand, for SM tracking, PTH₁R expression is kept low to achieve a low molecular density. In this context, FRAP of PTH$^{TMR}$ showed the slow diffusion of the PTH-PTH₁R complex at the ensemble level. This was supported by the similar mobile fraction obtained for both the active receptor and PTH$^{TMR}$ (Fig. 2I and Supplementary Fig. 7B). This led us to measure the diffusional properties of PTH in cells overexpressing the receptor. We anticipated that a high PTH₁R expression would favor direct association with the ligand, bypassing pre-association with the plasma membrane and showing increased lateral movement compared to cells with lower expression. The results shown in Supplementary Fig. 8 confirm this hypothesis.

Other PTH₁R agonists, such as the long- and short-acting PTH analogs, LA-PTH and M-PTH$_{1-14}$ respectively (Supplementary Table 1), induced a similar reduction in PTH₁R$^{mNG}$ diffusion (Supplementary Fig. 9). In contrast, the mobility of PTH₁R$^{mNG}$ remained unchanged in the presence of IA-PTH (Fig. 3D–G). Consequently, PTH₁R becomes trapped and reduces its diffusion when interacting with agonizts. Next, we characterized the dynamics of interacting trajectories between PTH₁R$^{mNG}$ and PTH$^{TMR}$ or IA-PTH$^{TMR}$ through two-color single-molecule coincidence detection within a distance <80 nm. This threshold was calculated from the MSD of fixed cells expressing PTH₁R$^{mNG}$ using a confined model to obtain the size of the confinement region[17]. The exact distance coincided with the standard deviation of the Gaussian fit from the point spread function of the fluorescent spots detected (Supplementary Fig. 10). Visual examination of interacting receptor–ligand trajectories revealed clear differences: PTH₁R$^{mNG}$ showed a limited lateral movement and confined mobility after its collision with PTH$^{TMR}$ but maintained its free diffusion when interacting with IA-PTH$^{TMR}$ (Fig. 3H, Supplementary Movies 2 and 3, and Supplementary Fig. 11). The lateral diffusion of interacting PTH agonizts (PTH, LA-PTH, and M-PTH$_{1-14}$) was significantly slower than that observed with the inverse agonist (Fig. 3I).

The calculation of diffusion coefficients using MSD analyzes a trajectory's complete path without considering the diffusion variations along the path followed by the receptor. We calculated the diffusion coefficient at the population level to address this limitation. This approach pooled displacements of trajectories at specific time intervals to determine heterogeneous behaviors of diffusion using cumulative distribution function (CDF, i.e., distribution of diffusion coefficient values)[18]. The CDF of radial displacements of PTH₁R$^{mNG}$ was best fit using a two-population model (Supplementary Fig. 12A). The fast population of PTH₁R$^{mNG}$ represented 67% of the displacements (Supplementary Fig. 12B) and exhibited a diffusion coefficient of $D = 0.14$ μm²/s with a Brownian motion ($\alpha = 0.92$) (Supplementary Fig. 12C–D). This fraction was significantly decreased to 45% and displayed slower diffusion with $D = 0.076$ μm²/s and a sub-diffusive motion ($\alpha = 0.7$) in response to PTH$^{TMR}$ stimulation (Supplementary Fig. 12C, D). In contrast, a predominant population (97%) represented PTH$^{TMR}$ motion, supporting the homogeneous behavior of the peptide ligand (Supplementary Fig. 12E, F). The diffusion coefficient obtained through MSD for PTH$^{TMR}$ was comparable to the slow population of PTH₁R$^{mNG}$, revealing similar values of diffusion and motion between the stimulated receptor and the ligand (Supplementary Fig. 12G, H). These results suggest that the slow population of PTH₁R$^{mNG}$ is interacting with PTH. Based on these findings, we propose that PTH binding to PTH₁R is initiated upon the lateral movement of the receptor and its random collision with the peptide ligand along the cell surface. PTH₁R diffusion is then reduced by full, but not inverse, agonists.

**Assembly of PTH–PTH₁R–βarr2 complex in pre-existing clathrin clusters.** The slow PTH₁R mobility when bound to PTH was not caused by its interaction with βarr2, given that similar diffusion coefficients were measured in HEK293 cells either lacking endogenous expression of βarr2 or expressing a previously characterized PTH₁R mutant unable to bind β-arrs (Thr-to-Ala mutation in residues 387 and 392 of PTH₁R referred to as PTH₁R$^{PD}$)[8] (Fig. 4A). Dual-color single-molecule microscopy of both PTH₁R$^{iRFP}$ and βarr2$^{mNG}$ further supported that the decrease of PTH₁R diffusion in response to PTH preceded its collision with βarr2 (Fig. 4B). We thus assume that βarr2 directly interacts with confined and slowly mobile PTH-bound receptors to assemble ternary PTH–PTH₁R–βarr complexes, which engage receptor internalization through either initiation of clathrin-coated pits (CCPs) or translocation of βarr-bound receptors into pre-existing CCPs[19–21]. To test this, we first determined the location of PTH-bound PTH₁R–βarr2 complexes by performing experiments at the single-molecule level in HEK293 cells endogenously expressing clathrin light-chain A (CLTA) alleles tagged with split mNeonGreen (CLTA$^{mNG}$)[18]. Here, dual-color single-molecule of PTH₁R$^{iRFP}$ and βarr2$^{PAmCherry}$ showed coincidental detection when both molecules were in CCPs (Fig. 4C). However, we found that PTH₁R activation did not alter the lifetime or the number of clathrin clusters, as assessed in CLTA$^{mNG}$ cells (Fig. 4D–G). Kymographs of CLTA$^{mNG}$ further supported the low lateral mobility of the CCPs, thus indicating that clathrin clusters do not diffuse to collide with receptors (Fig. 4D).

Next, we observed the appearance of PTH₁R–GFP (PTH₁R$^{GFP}$) in clathrin light chain Dsred (CLC$^{dsRed}$) spots through dual-color TIRF time-lapses (Fig. 4H). The increase of PTH₁R$^{GFP}$ fluorescence coincided with receptor location in preassembled CCP$^{dsRed}$, followed by a simultaneous decrease in both GFP and dsRed intensities, likely reflecting receptor internalization (Fig. 4I). Further, we found a delay in the plasma membrane recruitment of a βarr2 construct fused to dtTomato (βarr2$^{tom}$) to the plasma membrane, as compared to PTH₁R$^{GFP}$ recruitment in existing CCPs (Fig. 4G). These results suggest that the PTH-bound PTH₁R diffuses into pre-existing CCPs, where it assembles with βarr2 to form ternary PTH–PTH₁R–βarr2 complexes. We tested this hypothesis in the following experiments.

**PtdIns(3,4,5)P₃ control βarr2 recruitment**
We previously reported that PIP₃ is a critical regulator of βarr2 recruitment to the PTH₁R[7]. In this signaling model, the liberation of Gβγ from activated Gq in response to PTH promotes PI3Kβ-dependent generation of PIP₃, which in turn engages the formation and internalization of the ligand-bound PTH₁R–βarr2–Gβγ ternary complex. Here, we determined the role of PIP₃ on the initial process that forms the PTH₁R-βarr2 complex (Fig. 5A). First, we confirmed that PTH induces PIP₃ levels, using the biosensor containing tandem PH domains of ARNO fused to mCherry (aPHx2$^{mCherry}$) in HEK293 cells expressing the PTH₁R. This sensor is translocated to the plasma membrane upon increased PIP₃ production[22], measured as an increase in fluorescence at the plasma membrane. The PTH-mediated biosensor recruitment was entirely prevented by TGX-221, a selective inhibitor of the class-I phosphatidylinositol-3-kinase (PI3Kβ) (Fig. 5B).

Second, time courses recorded under TIRF illumination of FRET between PTH₁R fused to CFP (PTH₁R$^{CFP}$) and βarr2$^{YFP}$ confirmed that PIP₃ elevation mediated by PTH is required for the formation of the PTH₁R–βarr2 complex in response to PTH (Fig. 5C). Consistent with these observations, time course recordings in cells expressing βarr2 fused to YFP (βarr2$^{YFP}$) showed that βarr2$^{YFP}$ recruitment to the plasma membrane in response to PTH (10 nM) was blocked in the presence of TGX-221 (Fig. 5D). An implication from the results is that TGX-211 could block the process of PTH₁R internalization triggered by PTH. We examined this hypothesis using PTH₁R fused to super-ecliptic pHluorin, a pH-sensitive variant of GFP. Our findings indicate that PTH-activated PTH₁R internalized less and has a quicker recycling process in cells treated with TGX-221, or in cells lacking expression of β-arrestin isoforms 1 and 2 (Fig. 5E). Therefore, TGX-221 was not able to

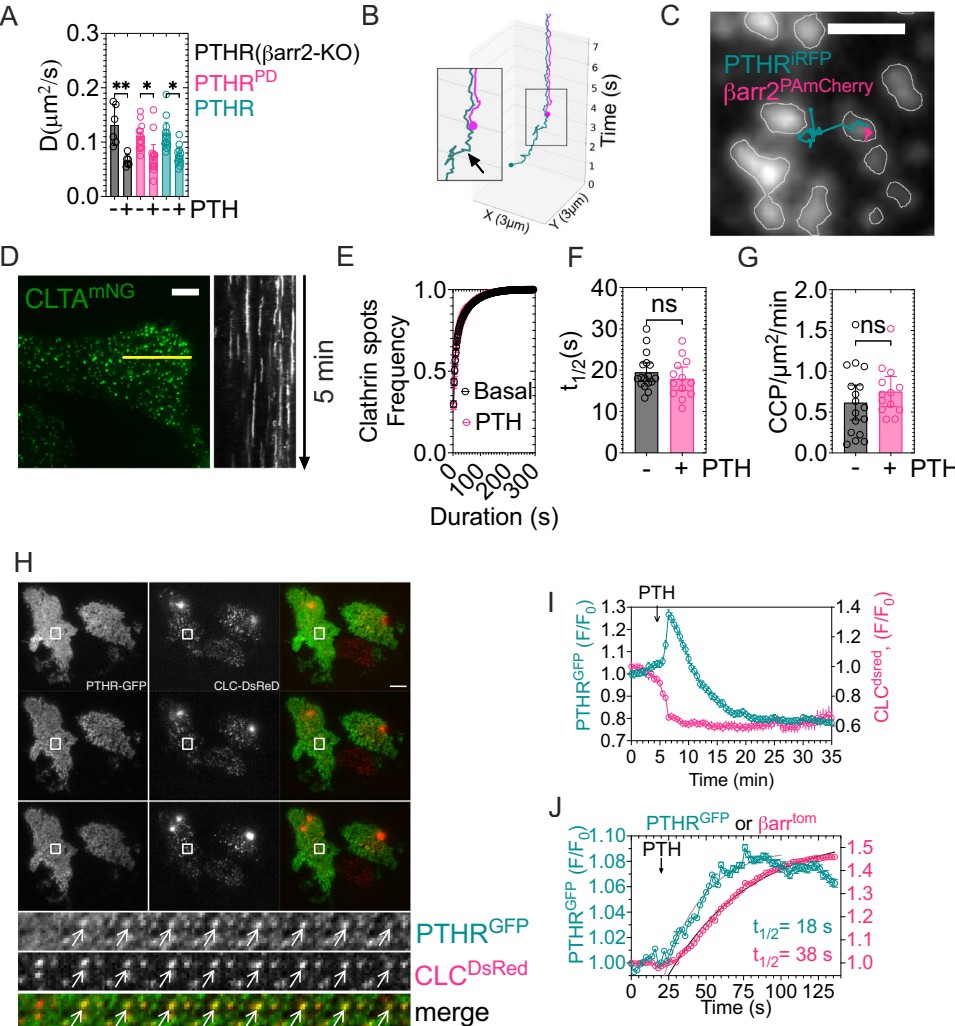

**Fig. 4 | PTH₁R in pre-existing CCPs. A** Diffusion coefficient values of PTH₁R^iRFP after PTH^TMR addition. Mean ± 95% C.I. of *n* (cells/average trajectories per cell) = 6/255 (PTH₁R^iRFP) and *n* = 6/464 (PTH₁R^iRFP + PTH) in βarr-KO cells, and *n* = 13/128 (PTH₁R^PD), *n* = 13/217 (PTH₁R^PD + PTH), *n* = 11/151 (PTH₁R^iRFP) and *n* = 11/175 (PTH₁R^iRFP + PTH) in control cells. **P < 0.01, *P < 0.05 by one-way ANOVA with a Dunnett multiple comparison. **B** Example of two-color single-molecule paths showing that the reduced PTH₁R^iRFP (teal) mobility occurs before its collision with βarr^mNG (magenta) as indicated by the arrow. **C** Single-molecule trajectories of PTH₁R^iRFP (teal) and βarr2^PAmCherry (magenta) in cells endogenously expressing mNG-labeled clathrin light chain A, CLTA^mNG (gray). Dots indicate the starting point. Scale bar, 1 μm. **D** Spatial distribution of CLTA (left) in CLTA^mNG cells, and kymograph representation of 5 min time-lapse of the yellow line of the cell on the right. Scale bar, 5 μm. **E**–**G** Cumulative frequency distribution of CCPs duration in cells expressing PTH₁R^iRFP in the presence or absence of PTH (**E**) and corresponding half-time values (**F**). Initiation rate of CCPs. The number of CCPs/min was normalized by the area of cell (**G**). Mean ± 95% C.I. of *n* = 17 (Basal) and *n* = 13 (PTH) cells of *N* = 4 experiments. ns, not significant by unpaired *t*-test. **H** PTH₁R clusters in pre-existing CCPs. HEK293 cells expressing PTH₁R^GFP and CLC^DsRed were stimulated with 100 nM PTH. Arrows indicate the formation of a PTH₁R^GFP cluster (upper line) in a pre-existing CCP (middle line). Scale bar, 10 μm. **I** Quantification from (**H**). Fluorescence levels from PTH₁R^GFP clusters and CLC^DsRed were normalized to the initial intensity. Mean ± SD of 4865 PTH₁R^GFP clusters from *n* = 18 independent cells. **J** Delayed βarr2 recruitment to PTH₁R. Cells expressing PTH₁R^GFP and βarr2^tom were stimulated with 100 nM PTH and imaged for 125 s. Fluorescence levels were normalized to the initial intensity. Mean ± SD of 3714 clusters from *n* = 18 independent cells.

entirely block PTH₁R internalization, which occurred via a βarr-independent internalization of PTH₁R.

Third, we further verified that the βarr2^YFP recruitment to the plasma membrane in response to PTH was blocked when PIP₃ was depleted from the plasma membrane by PTEN, the phosphatase that dephosphorylates PIP₃ to produce PtdIns(3,4)P₂[7,22,23]. Here we used an experiment previously optimized in HEK293 cells transiently expressing mcherry-tagged FK506 binding protein 12 fused to PTEN (mcherry- FKBP-PTEN), the FKBP12 rapamycin binding (FRB) domain of mTOR fused to Lyn11 (Lyn11-FRB), and the PIP₃ probe PH-Akt fused to venus (PH-Akt-venus)[7]. Rapamycin induces interaction between FKBP and FRP, bringing PTEN in contact with the plasma membrane (Fig. 6A). The inability of PTH to induce βarr2^YFP recruitment in the presence of rapamycin reinforced the critical role of PIP₃ in

translocating βarr2 to the plasma membrane by PTH-activated PTH₁R (Fig. 6B).

We confirmed these results at the single molecule level by showing that the number of membrane-bound βarr2^mNG molecules (Fig. 6C, D) and the time βarr2^mNG spent at the plasma membrane (referred to as dwell-time) were significantly (*P* = 0.0134) decreased in the presence of TGX-221 (Fig. 6E). Here, we measured the dwell-time in fixed cells expressing PTH₁R^mNG as control of mNG photobleaching at the plasma membrane. The shorter βarr2 dwell time measured in the presence of TGX-221 indicated that the absence of PIP₃, rather than fluorophore photobleaching, was rate-limiting (Fig. 6E). These results suggest that PIP₃ is a critical determinant in the recruitment and residence time of βarr2 molecules at the PM in response to PTH.

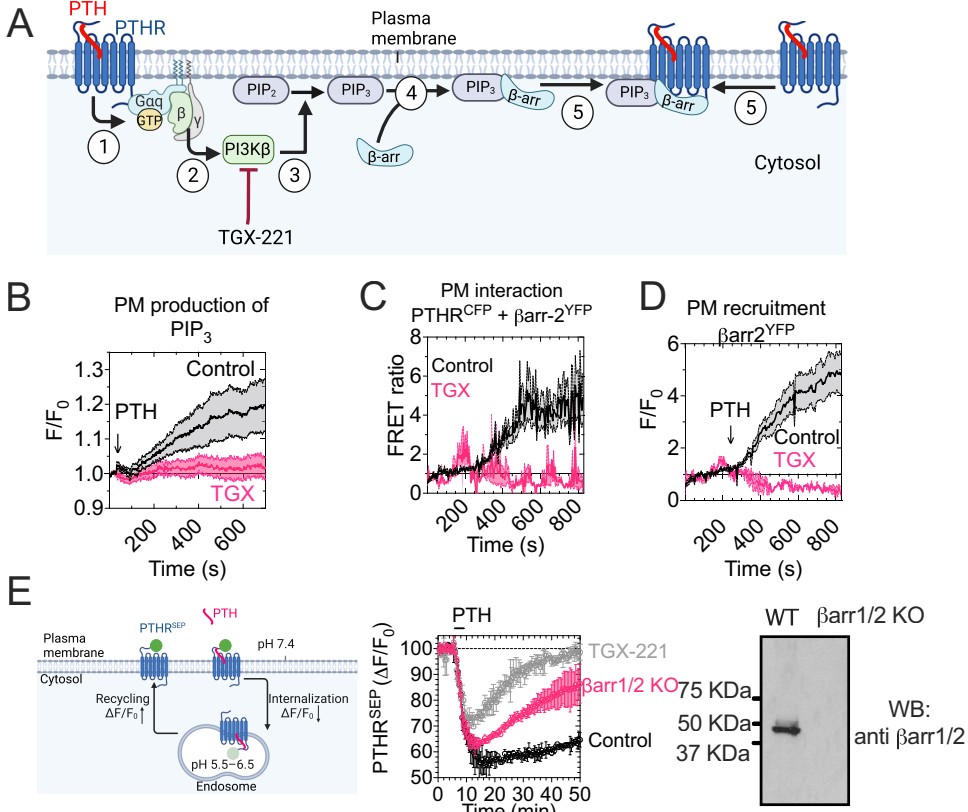

**Fig. 5 | PIP₃ controls βarr2 recruitment at the plasma membrane (part 1).**
**A** Principle of the experiments using TGX-221. PTH induces Gq activation (1); Gβγ-dependent recruitment and activation of PI3Kβ at the plasma membrane (2), which in turn catalyzes the formation of PIP₃ from PIP₂ (3); recruitment of βarr2 (4); and its interaction with PTH1R (5). TGX-221 is a potent PI3Kβ inhibitor. **B–D** Time-course recording under TIRF illumination of PIP₃ levels (**B**), FRET between βarr-2^YFP and PTH₁R^CFP (**C**), and βarr-2^YFP recruitment to the plasma membrane (**D**) in response to 10 nM PTH ± TGX-221 in HEK293 cells. Mean ± SEM of N = 3–6 experiments. **E** Time courses of internalization and recycling of the PTH₁R tagged with superecliptic pHluorin (PTHR^SEP) that exhibits fluorescence intensity reduction in acidic environments in response to PTH in HEK293 cells preincubated with DMSO (control, black) or 100 nM TGX-221 (pink) for 30 min before live cell imaging control and in βarr2-KO HEK293 cells, measured by time-lapse confocal microscopy. Cells were perfused with a 100 nM PTH (horizontal bar) and then washed out. Mean ± SD from N = 2–3 experiments and n ≥ 100 cells per condition. Expression of β-arrestins in parental and βarrs-KO HEK-293 cells using an anti-βarr1/2 antibody (*right* panel). Schematic in (**A**, **E**) created in BioRender. Vilardaga, J.P. (2025) https://BioRender.com/e81d105 and https://bioRender.com/o82q948, respectively.

## Direct interaction of PIP₃ and βarr2

We then tested whether PIP₃ can directly act on βarr2. Bead-based pull-down assays on separated cytosol and plasma membrane fractions of HEK293 cells expressing the recombinant HA-tagged PTH₁R (Fig. 7A) revealed that PIP₃ significantly (P < 0.0001) bound endogenous β-arrestins in the membrane fraction of stimulated cells (Fig. 7B, C). The enhanced capability of β-arrestin to attach to the PIP₃ resin when PTH is present suggests that the interaction between β-arrestins and PIP₃ is strengthened by the PTH-bound receptor or facilitated by post-translational modification of β-arrestins (such as phosphorylation) induced by PTH. Further insight into the action of PIP₃ on β-arrestin was obtained by testing PIP₃'s contribution to βarr2 conformation and activation. Given that the arrestin fingerloop has previously been shown to be essential for PTH₁R–βarr interaction[24], we used a purified minimal cysteine βarr2 variant with a cysteine residue replacing Leu69 in the fingerloop (L69C). We labeled it with bimane, a small environmentally sensitive fluorophore (βarr2-FL) (Fig. 7D, E and Supplementary Fig. 13)[25]. We compared the effect of PIP₃ with that of PtdIns(4,5)P₂ (henceforth PIP₂) and the phosphorylated C-terminal tail fragment of the vasopressin type 2 receptor (V2Rpp) as controls known to bind and induce conformational changes in β-arrestins[26]. Like PIP₂ and V2Rpp, PIP₃ induced a concentration-dependent increase in bimane fluorescence, suggesting a conformational change of the βarr2 consistent with activation (Fig. 7D, E). The higher affinity for PIP₃ indicates that it could be more effective for increasing and retaining

the pool of βarr2 at the plasma membrane than PIP₂ (Fig. 7F). While it was previously shown that βarr2 binds PIPs including PIP₃, via the C-lobe[27], we sought to confirm this binding promoted conformational changes analogous to that seen for βarr1[25]. We installed three additional mutations into βarr2 L69C, K233Q/R237Q/K251Q to compromise membrane PIP binding. While the mutant showed little difference in the conformational change elicited by the V2Rpp phosphopeptide, the conformational changes promoted by PIP₂ and PIP₃ were significantly diminished for the mutant (Fig. 7G). These results show that PIP₃ binds βarr2 with higher affinity than PIP₂ and induces a conformational change in βarr2 consistent with activation. These data support the idea that PIP₃ plays a functional role in the interaction between βarr2 and PTH₁R.

## Two-step process for βarr2 assembly with PTH-PTH₁R

Collectively, the findings support a two-step assembly process for the ternary PTH-PTH₁R-βarr2 complex. In the first step, βarr2 accumulates at the plasma membrane in response to PIP₃ elevation; in the second, βarr2 interacts with the PTH-bound receptor. To further support this two-step process, we assessed the time taken for individual βarr2^mNG molecules to collide with PTH₁R^iRFP once they have reached the plasma membrane. Using a coincidence detection of <80 nm (Fig. 8A), we simultaneously recorded single molecules of βarr2^mNG and PTH₁R^iRFP. Our analysis revealed a brief delay between βarr2 recruitment to the plasma membrane and its collision with PTH₁R. Histograms of pre-

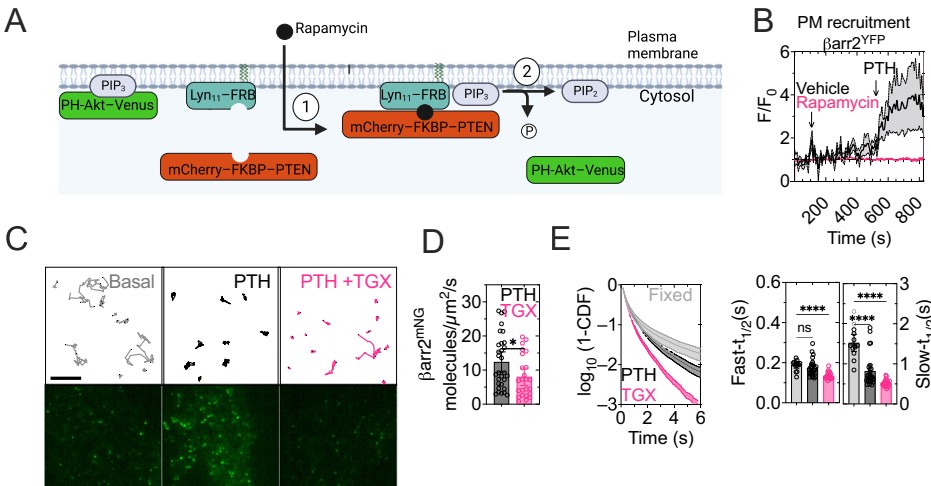

**Fig. 6 | PIP₃ controls βarr2 recruitment at the plasma membrane (part 2).**
**A** Principle of the experiment to recruit PTEN selectively to the plasma membrane through rapamycin-induced heterodimerization between FKBP (FK506 binding protein 12) and FRB (FKBP12 rapamycin binding) fused to mcherry-tagged PTEN (PTEN$^{mcherry}$) and Lyn₁₁, respectively; PIP₃ was detected using PH-Akt$^{venus}$. **B** Time-course recording βarr-2$^{YFP}$ recruitment to the plasma membrane under TIRF illumination in response to PTH (10 nM) ± rapamycin in HEK293 cells co-expressing PTH₁R, Lyn₁₁-FRB and FKBP-tagged PTEN$^{mCherry}$. Means ± SEM of $N = 6$–9 experiments. **C** Representative trajectories of βarr2$^{mNG}$ molecules in HEK293 cells expressing PTH₁R in the absence (basal), and the presence of PTH without (PTH) or with TGX221, (PTH + TGX221). Scale bar: 5 μm. **D, E** Average number of βarr2$^{mNG}$

molecules normalized by time (s) and the area of the cell in the presence of PTH alone or together with TGX-221 (**D**) and corresponding survival functions (1-CDF) for the duration of βarr2$^{mNG}$ trajectories (**E**, left). Curves were fit using a double exponential decay function (right). Mean ± 95% confidence intervals (C.I.) of $n = 14$/2744 (basal), $n = 27$/9099 (PTH), $n = 24$/6984 (PTH + TGX-221) and $n = 14$/5228 (fixed) (**D**) cells/average trajectories per cell of $N = 3$ experiments. *$P = 0.0311$ by unpaired $t$-test (**D**). ****$P < 0.0001$ and ns, not significant ($P = 0.3358$) by one-way ANOVA with a Dunnett multiple comparison with respect to fixed cells expressing PTH₁R$^{mNG}$ (**E**). Schematic in panel (**A**) created in BioRender. Vilardaga, J.P. (2025) https://BioRender.com/e81d105.

collision durations indicated that most collisions occurred rapidly, with a half-time delay ($t_{1/2}$) of 130 ms (Fig. 8B). A similar half-time value was calculated for the delay in βarr2$^{mNG}$ and PTH$^{TMR}$ colocalization (Fig. 8C).

Next, we examined the plasma membrane localization of individual molecules of PTH₁R$^{iRFP}$, PTH$^{TMR,}$ and βarr2$^{PamCherry}$ in HEK293A cells endogenously expressing CLTA$^{mNG18}$. The number of single-molecule detections was quantified through in-and-out binary masks from the fluorescent distribution of CLTA$^{mNG}$ by TIRF imaging (Fig. 8D-H). We found a similar probability of encountering PTH$^{TMR}$ in-or-out CCPs, indicating that the ligand was randomly distributed at the cell membrane (Fig. 8F). However, the time PTH$^{TMR}$ molecules spend at the plasma membrane increased inside CCPs only for cells expressing the PTH₁R, indicating that the complex ligand-bound receptor is stabilized on CCPs (Supplementary Fig. 14A, B). Similarly, the dwell-time of βarr2 molecules was significantly higher inside CCPs (Supplementary Fig. 14C). The PTH₁R$^{iRFP}$ did not show enrichment on CCPs before or after adding PTH or in cells treated with TGX-221 (Fig. 8G). However, the ratio of βarr2$^{PamCherry1}$ molecules in response to PTH was higher inside than outside CCPs, and it significantly decreased when cells were treated with TGX-221 (Fig. 8H). These results suggested that a significant fraction (~70%) of βarr2 molecules are associated with the PTH-PTH₁R complex in CCPs.

To investigate the remaining fraction of βarr2 not associated directly with CCPs (~30%), we asked whether PTH₁R, βarr2, and PTH can diffuse into pre-existing clathrin clusters separately or together in a complex. To address this point, we used a geometric approach. First, we used diffusion coefficient and motion values of PTH₁R$^{mNG}$ and βarr2$^{mNG}$ in basal condition (i.e., no PTH) and in complex (with PTH) using our dual-color interacting trajectories of either βarr2$^{mNG}$ and PTH$^{TMR}$ in cells expressing PTH₁R, or βarr2$^{mNG}$ and PTH₁R$^{iRFP}$ in the presence of PTH (Fig. 9A and Supplementary Table 3). Then, we generated binary masks from the distribution of clathrin spots to identify areas without clathrin clusters. Using this information, we constructed a Voronoi diagram to determine the radius of all empty circles

(Fig. 9B-D). Finally, we generated a plot reporting MSD vs. time-based on the calculated diffusion and anomalous motion values of PTH₁R and βarr2 freed of PTH and interacting PTH₁R, PTH, and βarr2. The results indicate that PTH₁R and βarr2 molecules explore an area of 6.9 μm²/min and 0.4 μm²/min, respectively, in the absence of PTH. If located in the center of an averaged empty clathrin surface of ~0.4 μm², PTH₁R and βarr2 can reach the border of a clathrin domain in ~1.1 and 56 s, respectively (Fig. 9E). However, the area of exploration of the PTH-PTH₁R-βarr2 complex dropped down to 0.03–0.06 μm²/min suggesting that the complex can only reach nearby clathrin clusters (Fig. 9E). These results indicate that the PTH-PTH₁R-βarr2 complex is more likely to assemble within clathrin clusters (Fig. 10).

## Discussion

The formation of a peptide hormone–receptor–βarr complex is an essential function of the cell membrane to transmit hormone actions. However, plasma membrane pathways and mechanisms used by peptides, receptors, and βarrs to assemble into a ternary complex are unknown. We have tackled this fundamental mechanism for the PTH₁R through a combination of single-molecule and biochemical analyses, resulting in two noteworthy findings.

The first addresses how PTH interacts with the PTH₁R at the single-molecule level. Earlier studies in lipid bilayers have proposed two major ligand-receptor interaction models[28,29], including (1) direct ligand-receptor interaction and (2) cell surface accumulation and lateral diffusion of ligands in the plasma membrane until they reach their receptor via lateral movements. This latter model proposes a reduction of dimensionality, first by 3-dimensional diffusion in the aqueous phase and then 2-dimensional diffusion in the plane of the cell surface until direct collision with an inactive receptor. Our results support an alternative model in which the receptor diffuses until its collision with the slow diffusing peptide ligand, serving as the initial ligand–receptor interaction step. Our results indicate that hyaluronic acid, a cell surface glycosaminoglycan (GAG), is likely responsible for PTH attachment at the plasma membrane. A recent study by our group reported the

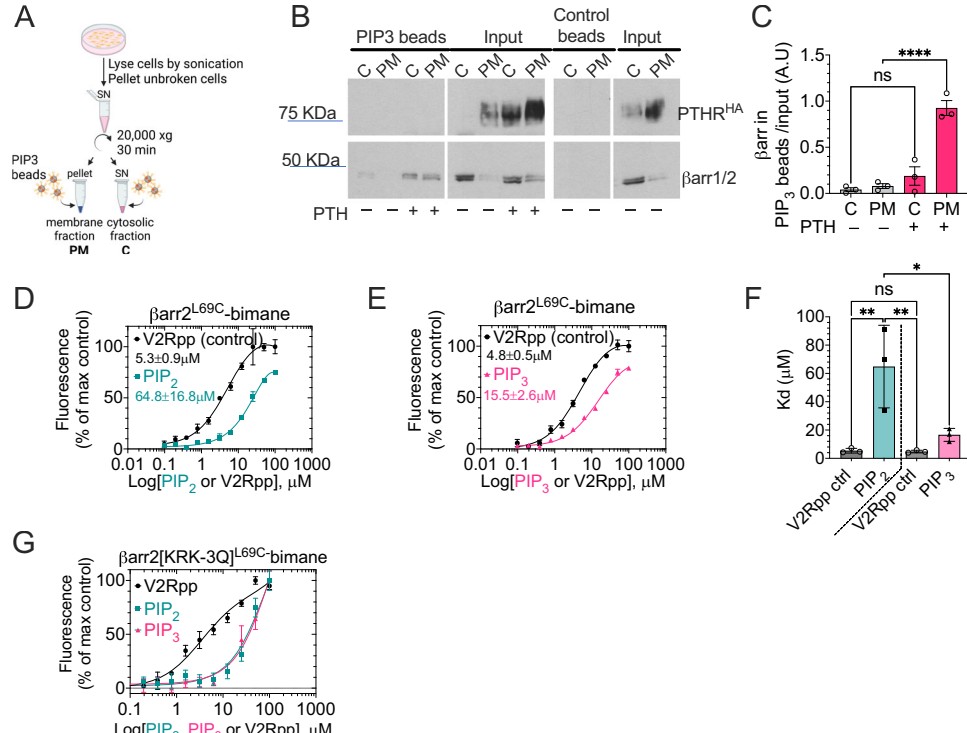

**Fig. 7 | βarr2 and PIP$_3$ interaction. A–C** Principle of the PIP$_3$ bead-based pull-down assay (**A**). Immunoblot analysis (**B**) and quantification (**C**) of β-arrestins binding to PIP$_3$ in HEK293 cells stably expressing the HA-tagged PTH$_1$R. Cytosolic proteins and solubilized membrane proteins were separated by SDS-PAGE. βarr isoforms were visualized by immunoblot analysis with a βarr1/2 antibody (**B**). Mean ± SEM of $n = 3$ experiments. ****$P < 0.0001$ by one-way ANOVA with a Tukey's multiple comparison. Schematic created in BioRender. Pena, K. (2024) https://BioRender.com/

g93j700. **D–F** Concentration-response curves of PIP$_2$ (**D**), PIP$_3$ (**E**), and V2Rpp for βarr2L69C-BIM, and corresponding Kd values (**F**). *Y*-axes, ΔF/F shows fluorescence intensity (at λmax. see Supplementary Fig. 13). Means ± SEM of $n = 3$ experiments. *$P < 0.03$, and **$P < 0.002$ by one-way ANOVA with a Tukey's multiple comparison. **G** Concentration-response curves of PIP$_2$, PIP$_3$, and V2Rpp for the KRK-3Q mutant of βarr2L69C-BIM. Mean ± SEM of $n = 3$ experiments.

dependence of cAMP signaling mediated by the native PTH-related protein (PTHrP$_{1-141}$) on heparin, a decoy for cell surface[30]. Whether additional GAGs could serve as possible means to anchor PTH to the cell surface remains to be verified. Additionally, the role of CCPs as sinks of PTH in the presence of PTH$_1$R should be considered. More research is required to determine the exact mechanism by which the cell surface confines PTH and PTHrP, and whether such a mechanism can be extended to other peptide hormones.

The second explains how the ternary PTH–PTH$_1$R–βarr2 complex is assembled. The assembly involves a sequence of steps and molecular interactions at the plasma membrane that starts when PTH$_1$R collides and binds PTH via lateral diffusion. This collision can occur outside or inside clathrin spots, given that PTH is slowly diffusing and randomly distributed with ≈50% of molecules encountered in clathrin clusters. Here, we showed that PTH$_1$R activation does not induce the formation of new clathrin clusters, and the PTH–PTH$_1$R–βarr2 complex takes place on pre-formed CCPs (Fig. 4). This process has been reported for other GPCRs, such as the class-B thyrotropin-releasing hormone receptor (TRHR)[21] and the class-A β$_2$-adrenergic receptor (β$_2$AR)[31,32].

The PTH-bound receptor induces activation of Gq and PI3Kβ, producing PIP$_3$, which stabilizes the βarr2 contact with the PTH$_1$R. Biochemical results reported in Fig. 7 and the delay before βarr2 and PTH$_1$R collisions observed in Fig. 8 suggest that PIP$_3$ may promote conformational rearrangements in βarr2 before interacting with the PTH-bound receptor. Then, βarr2 collides with the PTH-bound receptor through lateral diffusion. The recent study by Grimes et al.[33], provides insights into the sequential steps of βarr2 before interacting with the β$_2$AR in clathrin-coated pits. Our work, in alignment with Grimes et al., reveals the spontaneous binding of βarr2 to the plasma membrane before receptor activation, followed by a reduction of diffusion and an increase of dwell time at the plasma membrane after receptor activation. Specifically, in the case of the β$_2$AR, βarr2 initially binds to the plasma membrane and adopts its active conformation through random collisions with active receptors. This process necessitates βarr2's lateral diffusion outside of CCPs before accumulating in CCPs, leading to the separate lateral diffusion of βarr2 and β$_2$AR to CCPs[33]. On the other hand, our observation after PTH$_1$R activation reveals a different scenario. We noticed a limited diffusion of βarr2, challenging lateral exploration outside CCPs. However, βarr2 reaches the plasma membrane directly in areas with pre-formed CCPs, where it performs a brief exploration before colliding with the complex PTH-PTH$_1$R. The role of PIP$_3$ here is pivotal, as a significant fraction (≈70%) of βarr2 molecules are recruited to CCPs in dependence on PIP$_3$ (Fig. 6). The class of GPCR could explain these differences on βarr2, as Grimes et. al. showed for chimeras GPCRs that increase affinity for βarr2 favor association inside CCPs. Based on the predictive kinetics of βarr2 diffusing into CCPs shown in Fig. 9, the accumulation of the PTH-PTH$_1$R-βarr complex in clathrin clusters is slower than βarr2 diffusing into pre-existing clathrin clusters separately from the PTH-PTH$_1$R complex (Fig. 10).

In conclusion, our findings suggest that the ligand-bound receptor is formed after receptor collision with the slow diffusing peptide ligand. Our results support a model where βarr2 recruitment to the plasma membrane and its interaction with PTH$_1$R requires PIP$_3$ and clathrin clustering. The mechanism of PTH-trapping via GAGs would imply that cells can assemble activated receptors in specific areas of the cell surface. As the PTH$_1$R-βarr is more effectively assembled in clathrin domains, this mechanism would favor receptor internalization and subsequent endosomal PTH$_1$R signaling. Further research is

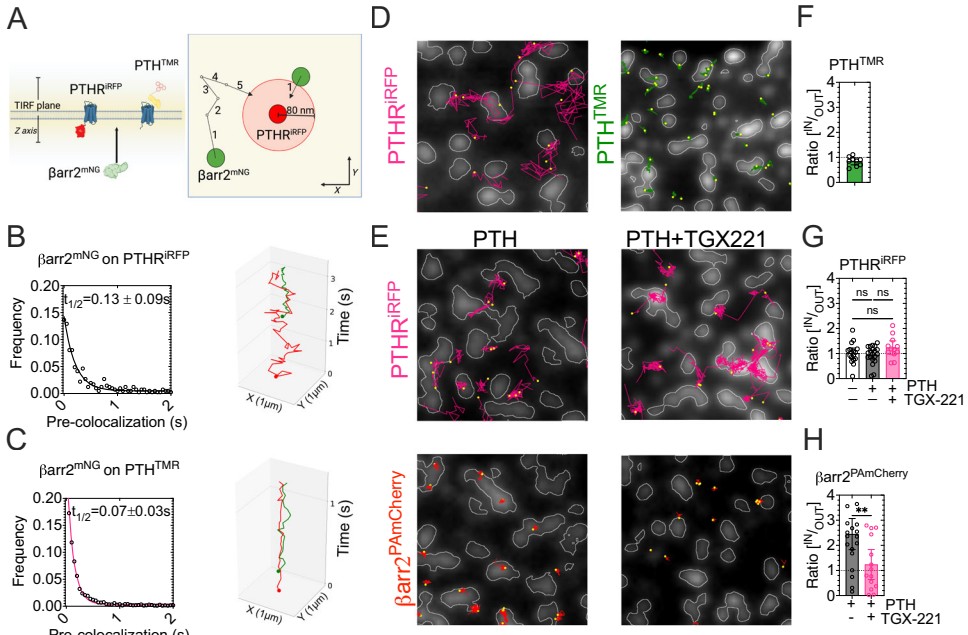

**Fig. 8 | Two-step process for βarr2-PTH₁R association. A** Principle of the analysis. Dual-color single-molecule was performed under TIRF illumination with βarr2$^{mNG}$ and PTH₁R$^{iRFP}$ (**B**) or PTH$^{TMR}$ in cells expressing PTH₁R$^{CFP}$(**C**). Under the assumption that all molecules of βarr2$^{mNG}$ detected on the TIRF plane are on the plasma membrane. The duration of βarr2$^{mNG}$ molecules before their coincidental detection (<80 nm) with a receptor was quantified; this "pre-colocalization time" reflects the time βarr2$^{mNG}$ spent free at the PM before collision with PTH₁R$^{iRFP}$. Scheme created in BioRender. Pacheco Romero, J. (2025) https://BioRender.com/t93c242. **B, C** Pre-colocalization time-frequency distributions for βarr2$^{mNG}$ interacting with PTH₁R$^{iRFP}$ (**B**) or PTH$^{TMR}$ (**C**) (left panels). The data was fit with a single exponential decay. Histograms were produced from pooled data derived from $n = 7/61$(βarr2$^{PAmCherry}$/ PTH₁R$^{iRFP}$) and $n = 7/568$ (βarr2$^{PAmCherry}$/PTH$^{TMR}$) cells/average events per cell in 3 experiments. 3D plot of trajectories showing fast association of βarr2$^{mNG}$ (green) with PTH₁R$^{iRFP}$ (red) (**B**, right panel) or PTH$^{TMR}$ (red) (**C**, right panel). Dots in

trajectories indicate the start point. **D, E** Distribution of PTH₁R$^{iRFP}$, PTH$^{TMR,}$ and βarr2$^{PAmCherry}$ trajectories on endogenous clathrin clusters (clathrin light-chain A, CLTA$^{mNG}$). Examples showing the fluorescent distribution of CLTA$^{mNG}$ and the resulting binary mask (white line contour). Magenta, green, and red trajectories represent single molecule tracking of PTH₁R$^{iRFP}$, PTH$^{TMR}$, and βarr2$^{PAmCherry}$, respectively. In panel (**E**), cells were treated with PTH or PTH + TGX221. **F–H** The ratio of single-molecule detected inside over outside clathrin-domains normalized by the area of the cell. Data are the mean ± 95% confidence intervals (C.I.). For PTH₁R$^{iRFP}$, $n = 16/26738$ (basal), $n = 20/26637$ (PTH), $n = 13/39071$ (PTH + TGX221) cells/average detections per cell of $N = 3$ experiments. ns, not significant by one-way ANOVA with a Tukey's multiple comparison (**G**). For PTH$^{TMR}$, $n = 9/436960$ from $N = 3$ independent experiments and for βarr2$^{PAmCherry}$ $n = 20/14050$ (PTH) and $n = 15/10929$ (PTH + TGX221) of $N = 4$ and 5 experiments, respectively. **\*\*$P = 0.0074$** by unpaired *t*-test (**H**).

---

needed to determine if this mechanism can be applied to other peptide ligand GPCRs.

## Methods

### Cell culture and transfection

Cell culture reagents were obtained from Corning (CellGro). Human embryonic kidney (HEK293; American Type Culture Collection (ATCC), Georgetown, DC) cells were grown in DMEM (low glucose; Life Technologies 10567022) supplemented with 5% heat-inactivated fetal bovine serum (Life Technologies 10438-034), 100 units/ml penicillin, 100 μg/mL streptomycin (Life Technologies 15140122) at 37˚C in a humidified atmosphere with 5% CO2. Human renal epithelial tubular epithelial cells immortalized with hTERT (RPTEC generously provided by Dr. Peter A. Friedman are from ATCC, Georgetown, DC) were grown in DMEM/F12 supplemented with 5 pm triiodo-L-thyronine, 10 ng/ml recombinant human epidermal growth factor, 25 ng/ml prostaglandin E1, 3.5 μg/ml ascorbic acid, 1 mg/ml insulin, 0.55 mg/ml transferrin, 0.5 μg/ml sodium selenite, 25 ng/ml hydrocortisone, plus 1% penicillin and streptomycin. Mouse osteoblast-like MC3T3-E1 cells, subclone 4, (MC3T3, ATCC, CRL-2593) were grown in α-MEM supplemented with 10% fetal bovine serum (FBS). HEK293A cells expressing endogenous clathrin light-chain A (CLTA) tagged with split mNeonGreen were previously described[18,34]. Signaling studies were performed on cells stably expressing the recombinant human HA-tagged PTH₁R[35]. For single-molecule imaging studies, cells were seeded in 35 mm tissue culture dishes with 20 mm number 1.5 cover glass apertures (CellVis) previously

treated with 10 μg/mL of human fibronectin protein (Thermo Fisher Scientific 33016015) for 30 min at 37˚C. Cells were transfected ~24 h post-seeding using 3 μg Lipofectamine 2000 (Life Technologies 11668019) in 200 μL Opti-MEM (Life Technologies 51985091) per dish. 0.2 μg of plasmid containing PTH₁R was used to make receptor expression compatible for single molecule recording. For cells co-expressing PTH₁R and β-arrestin2, 0.2 μg and 0.8 μg of each plasmid were transfected, respectively. Cells were imaged 4 h post-transfection.

### Chemicals and reagents

Human PTH$_{1–34}$ was purchased from Bachem. TMR-labeled peptides are listed in Supplementary Table 1. PTH$^{TMR}$ was purchased from Life-Tein LLC; PTH$_{1-34}$$^{fluorescein}$ and TMR-labeled LA-PTH, M-PTH$_{1-14}$, and PTHrP$_{7-36}$ were synthesized by the MGH peptide core facility and generously provided by Tom Gardella and Ashok Khatri (Endocrine Unit, Massachusetts General Hospital and Harvard Medical School). Peptides were resuspended in 10 mM acetic acid to make 1 mM stock solutions. Forskolin (#344270) was purchased from EMD-Millipore. PTH was used at 10 and 50 nM for PTH1R and β-arrestin 2 experiments, respectively. TGX-221 purchased from MedChemExpress LLC was dissolved in DMSO at 100 mM, stored as a stock at −20˚C, and used in cells at 100 nM for 30 min before imaging.

### Plasmids

All plasmids were constructed using NEB HiFi assembly or standard DNA recombinant techniques. PTH₁R and β-arrestin2 were previously

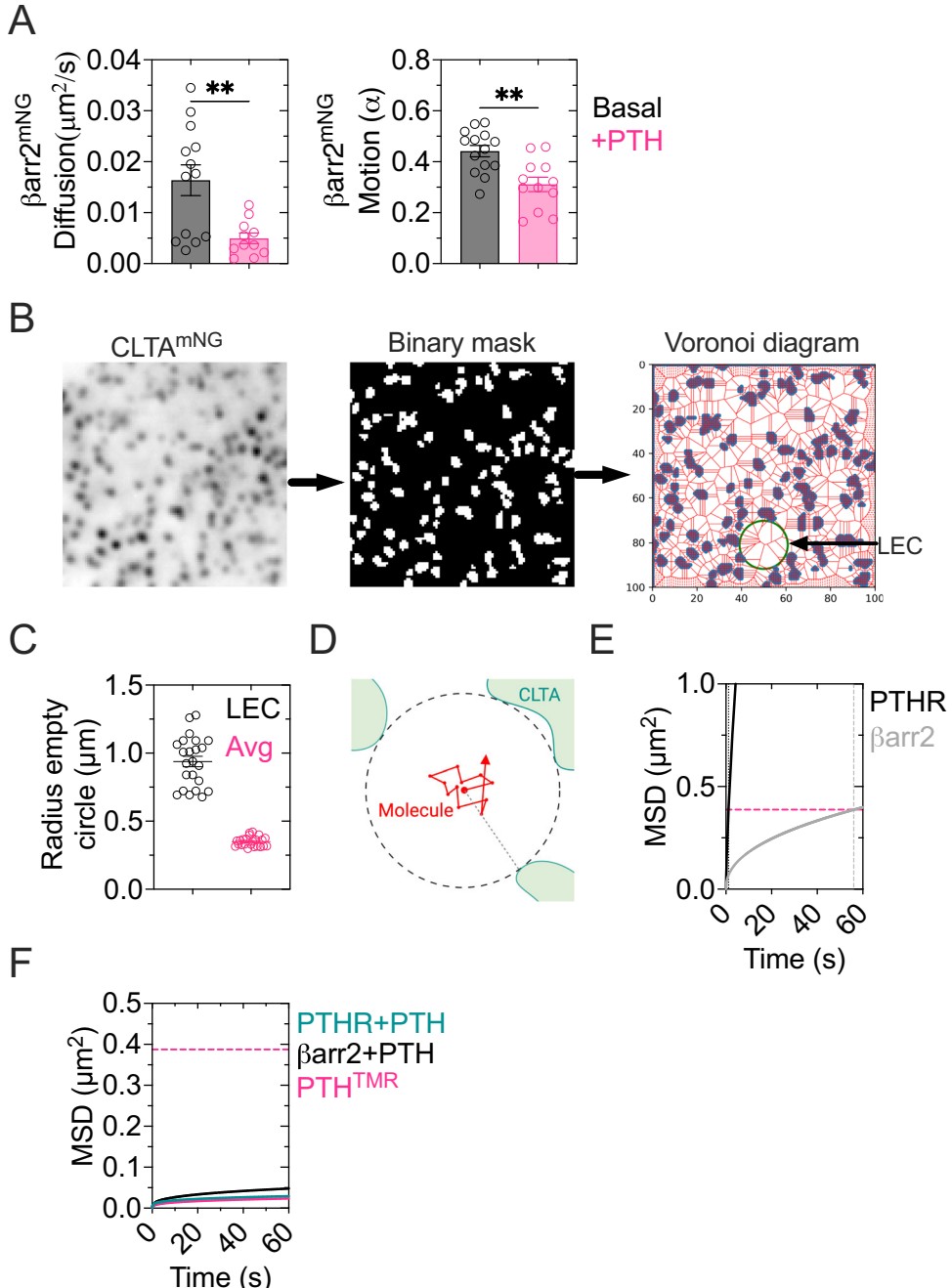

**Fig. 9 | βarr2 diffusion on CCPs domains. A** Diffusion coefficient and motion values of βarr2[mNG] trajectories in the absence and presence of PTH. Data is the mean ± 95% confidence intervals (C.I.) of $n = 14/1093$ (basal) and $n = 26/1032$ (PTH) cells/average trajectories of at least $N = 3$ experiments. **$P = 0.0074$ by unpaired $t$-test. **B–D** Scheme illustrating the process of calculating the clathrin-free areas. First, a binary mask was produced from the distribution of CLTA[mNG]. A Voronoi tessellation diagram was generated from this binary mask. The convergence of vertices in the diagram corresponds to the center of a circle with a radius equal to the distance from a point centered on the farthest edge of a clathrin border (**B**). The area of the largest empty circle (LEC) and the average of all empty circles were computed (AEC) (**C**). Data are the mean ± S.E.M. For $n = 23/933$ cells/average empty circles per cell in 3 independent experiments. Scheme showing a hypothetical molecule recruited to the plasma membrane in the center of a clathrin-empty circle (**D**). Scheme created in https://BioRender.com. **E**, **F** Model of MSD *vs* time, generated from diffusion values of trajectories for PTH₁R and βarr2 in the absence of PTH (**E**), and PTH₁R, βarr2 in presence of PTH and PTH[TMR] (**F**). The horizontal discontinued magenta line represents the area of the average empty circles (AEC). The solid lines show when βarr2 and PTH₁R (**E**, **F**) would reach a clathrin spot in the absence (**E**) or presence (**F**) of PTH.

described[4]. pPAmCherry1-N1 (Addgene plasmid 31929) and iRFP670 were reported in ref. 22. mNeonGreen was described in ref. 36. Sources and backbones are indicated in Supplementary Table 2. Indicated plasmids were under a truncated CMV promotor containing the last 155 base pairs of CMV. All plasmids were verified by dideoxy sequencing.

## Single-molecule microscopy

Cells were imaged in 2 ml of pre-warmed (37 °C) FluoroBrite DMEM (Life technologies A1896702) supplemented with 25 mM Hepes (pH 7.4) and 5% heat-inactivated fetal bovine serum. Before imaging, cells were washed once with 1 ml of FluoroBrite DMEM. Single-molecule imaging experiments were performed on a Nikon TiE motorized

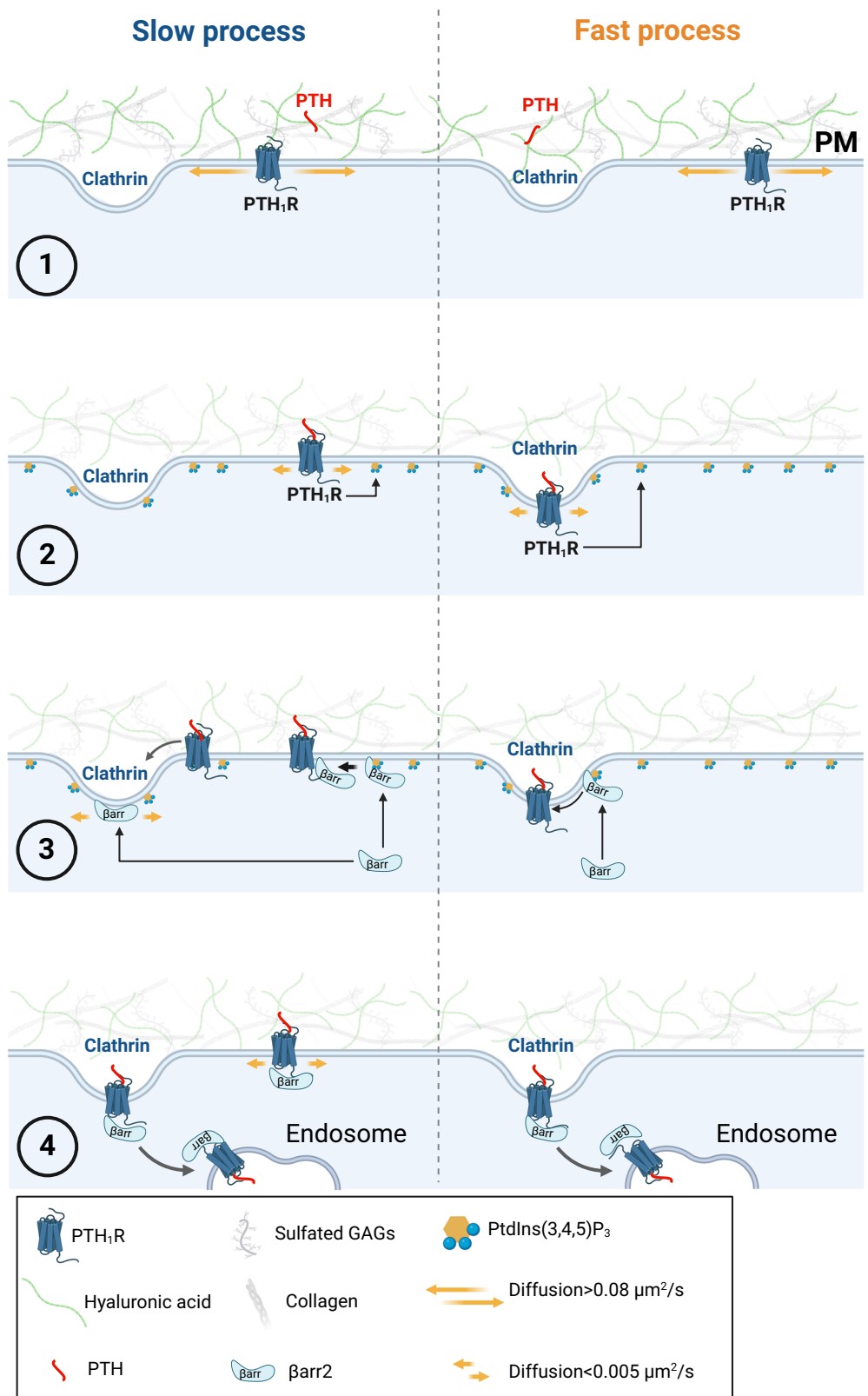

**Fig. 10 | Proposed model of the PTH-PTH1R-βarr2 complex assembly.** The receptor diffuses with near-Brownian motion until its collision with PTH which is temporarily retained at the cell surface by hyaluronic acid and randomly distributed (step 1). The PTH-bound-receptor has reduced diffusion and motion and induces PIP3 production at the plasma membrane (step 2), which in turn recruits βarr2 to the plasma membrane (step 3). The accumulation of the PTH-PTH1R-βarr2 complex in clathrin clusters can follow different pathways: a fast pathway where the larger accumulation of PIP3-recruited βarr2 in clathrin clusters interacts with the PTH-bound PTH1R in clathrin (steps 3 and 4, *right*); a slower pathway where βarr2 and PTH-PTH1R diffuse either as a complex or separately into nearby (<100 nm) clathrin spots (steps 3 and 4, *left*). Created in BioRender. Pacheco Romero, J. (2025) https://BioRender.com/u62m09.

inverted microscope equipped with a TIRF illuminator with a 100X, 1.45 NA plan-apochromatic oil-immersion objective. The illumination source was an Oxxius L4C launch with 405, 488, 561, and 638 nm lasers. An ORCA-Fusion BT sCMOS camera (Hamamatsu) was used for single-molecule detection. Images were captured with a 2 × 2 binning, a frame size of 250 × 250 pixels, and 1.5X optical magnification to give a pixel size of 85 nm. Frame exposure was set at 50 or 25 ms with the ultra-quiet and fast mode of the camera for single and two-color single-molecule recordings, respectively. Nikon Elements software was used to control the microscope, and image data was saved as ND2 files. For single-molecule experiments in the upper membrane (Fig. 2), transfected cells were seeded on 18 mm dishes, previously treated with fibronectin, and incubated with 10 nM PTH$^{TMR}$ for 30 s. Afterward, the cells were washed three times with a fresh Fluorobrite medium. The coverslip was then carefully positioned on the piston of the adapter. The coverslip was inverted and placed on an uncoated 35 mm tissue culture dish with 20 mm number 1.5 cover glass apertures.

A sequential acquisition was used under a triggered mode controlled by NIS-Elements software for two-color single-molecule experiments. This configuration allowed us to alternate between lasers 488 and 561 nm (PTH$_1$R$^{mNG}$/PTH$^{TMR}$) or 488 and 638 nm (βarr2$^{mNG}$/PTH$_1$R$^{iRFP}$) with a minimal delay of 25 ms between channels. Laser power was 20%, 30%, and 100% for 488, 561, and 638 nm lasers respectively. For two-color single-molecule experiments PTH$_1$R$^{mNG}$/PTH$^{TMR}$, 10 nM of ligand was added in live cell recordings, allowing 30 s before stimulation as a basal condition. For βarr2$^{mNG}$/PTH$_1$R$^{iRFP}$ experiments, cells were stimulated with PTH (50 nM). Two-color time-lapses were recorded for 5 min after PTH stimulation. One color single-molecule recordings occurred from 3 to 15 min after stimulation with PTH, these recordings were processed to calculate the diffusion coefficient of PTH$_1$R$^{mNG}$ and the dwell-time of βarr2$^{mNG}$ (see next sections).

### Single-molecule tracking

Time-lapse recordings were opened on Fiji software using the Bio-formats importer plugin. Trajectories were generated using the TrackMate plugin[37,38], with a difference of Gaussians filter and 0.5 μm of estimated diameter for detection. The same distance was used for linking and gap-closing by implementing a simple linear assignment problem algorithm. The maximal distance used was 0.6 and 0.3 μm for PTH$_1$R and βarr2/PTH$^{TMR}$ respectively. The coordinates of trajectories were exported as.csv files and analyzed with custom codes in Python 3.9.7.

### Determination of diffusion coefficient values

To calculate the apparent diffusion coefficient from mean-square displacements only the trajectories longer or equal than 0.8 s were considered for the analysis. This size corresponds to 16 frames. All trajectories >0.8 s were cut at 16 frames to homogenize trajectory sections. The mean-square displacement (MSD) was calculated from all pairs of points[39]:

$$\langle r^2(i\Delta t)\rangle = \frac{1}{N_A}\sum_{j=1}^{N_A-1}\left[\vec{r}[(j+i)\Delta t] - \vec{r}(j(\Delta t)]\right]^2 \qquad (1)$$

The apparent diffusion coefficient was calculated from MSD as function of the first 4-time intervals[40]:

$$MSD_{1-4} = 4Dt^\alpha \qquad (2)$$

The exponent alpha (α) denotes the anomalous motion. For Brownian motion, 1 < α < 0.7. For subdiffusive motion, α < 0.7. Unless otherwise indicated, the grand mean of the median of all trajectories per cell is plotted. Diffusion coefficient values were also estimated

from radial displacements using cumulative distribution functions (CDF) from 1-to-4 time intervals in all trajectories >0.2 s. Each histogram corresponding to a particular time interval (iΔt) was fit with a 2-population model[41]:

$$P_{(r, i\Delta t)} = 1 - \left(e^{-r^2/4D_{fast}(i\Delta t)} + e^{-r^2/4D_{slow}(i\Delta t)}\right) \qquad (3)$$

where r is the radial displacements at iΔt, D$_{fast}$ and D$_{slow}$ are diffusion for fast and slow fraction respectively. The anomalous motion was calculated from the change of diffusion along the first 4-time intervals:

$$D = D_n t^{\alpha-1} \qquad (4)$$

where D$_n$ is D$_{fast}$ or D$_{slow}$ and the exponent α is the anomalous motion for each population.

Given the observation of a certain amount of background PTH$^{TMR}$ molecules bound to the glass-coverslip in the absence of cells, a correction of diffusion values was performed to ensure accurate analysis. The contribution of these background molecules to the molecules of interest was subtracted through the following steps: (1) the diffusion coefficient of PTH$^{TMR}$ molecules on coverslips coated with fibronectin was measured. This measurement established a baseline diffusion coefficient for the background molecules. The baseline diffusion coefficient was determined by calculating the median diffusion for MSD analysis, the first four-time intervals (0.05 s, 0.1 s, 0.15 s, and 0.2 s), or the radial displacement analysis. The cut-off value for distinguishing background diffusion was set as the median diffusion plus two standard deviations (0.00082 μm$^2$/s for MSD). Using these thresholds, molecules with a diffusion coefficient above the calculated threshold were considered molecules of interest, while those below the threshold were classified as background molecules and discarded from the analysis. (2) to ensure diffusion measurements specifically on PTH$^{TMR}$ molecules attached to the membrane, another level of control was implemented by measuring only trajectories that exhibited coincidental detection with PTH$_1$R, as confirmed by two-color experiments.

### Precision measurements in fixed cells

Fixed cells were used as control of precision measurements to establish a threshold distance in coincidental detection in dual-color single-molecule experiments, and as a control of photobleaching in dwell-time measurements. Cells expressing PTH$_1$R$^{mNG}$ were fixed with 0.2% Glutaraldehyde dissolved in Phosphate-buffered saline (PBS) for 15 min. Cells were washed twice with freshly made 10 mg/ml Sodium Borohydride dissolved in PBS. Cells were imaged in Fluorobrite medium. A threshold distance of 80 nm was calculated through the localization accuracy of fixed cells. Given that the dynamics of PTH$_1$R$^{mNG}$ on fixed cells showed an immobile diffusion, we defined the size of confinement as localization accuracy. For this, the average MSD was fit by using the first 4-time intervals. MSD curves were fit with an equation for confined diffusion[42]:

$$MSD = \left(\frac{L_i^2}{3}\right)\left(1 - e^{-(\Delta t/\tau_i)}\right) + offset \qquad (5)$$

where L$_i$ is the size of confinement, τ$_i$ is the equilibration time. The calculated localization accuracy does match with one standard deviation of the particle size, calculated through ThunderSTORM plugin (~87 nm).

### Single molecule colocalization

The time of colocalization was measured by defining 80 nm as the threshold distance of coincidental detection in the same corresponding frame. All the coordinate positions of trajectories in channel-1 were compared with channel-2 to find colocalizing molecules at the threshold distance in adjacent frames. For this analysis, only

trajectories >0.2 s were considered. The optimization of colocalizing trajectories was carried by removing colocalizing events of one frame duration and allowing the interruption of coincidental detection by up to 2 frames.

## Clathrin enrichment

To study β-arrestin dynamics inside of clathrin clusters, we made a sequential acquisition of βarr2[PAmCherry] in HEK293A cells endogenously expressing clathrin light chain A (CLTA) fused to split mNG. Clathrin fluorescence distribution was used to generate a binary mask through à trous wavelet decomposition algorithm[43] as previously described[44]. Recordings of 30 s were generated by switching 488 nm and 561 nm lasers with a 1% constant 405 nm laser to photoactivate PAmcherry fluorescence. Individual detections falling inside the binary masks were normalized by the area of the cell using the footprint of the clathrin signal and compared with the number of localizations outside of clathrin domains. The data is presented as a ratio of localizations inside clathrin divided by the localizations outside clathrin. For the calculation of all empty circles and the largest empty circles, we utilized the same binary masks of CLTA cells. A triangulation protocol was employed on all clathrin borders to identify the circumcenter distances in areas devoid of clathrin. The radii of all empty clathrin areas were computed to extract the average (AEC) and the largest (LEC) empty circle. All edges on the frame were discarded to remove distances not covering the real clathrin domains. Subsequently, we simulated the diffusion coefficient through mean square displacement (MSD) to calculate the time required for PTH$_1$R, β-arrestin and PTH to reach the area corresponding to a clathrin border.

## Fluorescence recovery after photobleaching

FRAP experiments were performed on a Nikon TiE A1R confocal with a resonant mode and a 100 × 1.45 NA plan-apochromatic objective. Cells expressing 1 μg of PTH$_1$R[mNG] were incubated 10 nM PTH[TMR] for 30 s. Fluorobrite medium was replaced after ligand incubation. Time courses were set with a pinhole of 0.5 AU, a frame size of 1024 × 1024 pixels, and a pixel size of 0.082 μm following the Nyquist criteria. A sequential acquisition between green and red channels were acquired each 250 ms by using 2 and 5% power of 488 nm and 561 nm lasers, respectively. A total of 5 min were recorded allowing 10 s before photobleaching of a 11 μm diameter circular region-of-interest (ROI) for 20 s with 100% power of 405 nm laser.

For FRAP experiments in a TIRF setup, cells expressing PTH$_1$R[mNG] and labeled with PTH[TMR] were imaged before and after photobleaching the entire frame using lasers at 405 nm, 488 nm, and 561 nm with maximum intensity for 10 s. Time courses were recorded for 5 min. The mobile fraction was calculated as the percentage of fluorescence recovery after 250 s of photobleaching.

## Time-course measurements of cAMP production, and PTH$_1$R recruitment of β-arrestin 2 in single live cells

Cyclic AMP and β-arrestin 2 recruitment were assessed using single-cell FRET-based assays previously described[45]. In brief, cells were transiently transfected with the Epac1-CFP/YFP for measuring cAMP production, and PTH$_1$R C-terminally fused to CFP (PTH$_1$R-CFP) with βarr2-YFP for measuring arrestin recruitment. Measurements were performed on cells plated on poly-D-lysine coated glass coverslips and mounted in Attofluor cell chambers (Life Technologies) and maintained in HEPES buffer containing 150 mM NaCl, 20 mM Hepes, 2.5 mM KCl and 1 mM CaCl$_2$, 0.1% BSA, pH 7.4. Cells were imaged on a Nikon Ti-E motorized inverted microscope with Perfect Focus System and equipped with an oil immersion 40X N.A 1.30 Plan Apo objective and a moving stage (Nikon Corporation). CFP and YFP were excited using a mercury lamp. Fluorescence emissions were filtered using a 480 ± 20 nm (for CFP) and 535 ± 15 nm (for YFP) filter set and collected simultaneously with a LUCAS EMCCD camera (Andor Technology)

using a DualView 2 (Photometrics) with a beam splitter dichroic long pass (DCLP) of 505 nm. Fluorescence data were recorded from single cell using Nikon Element Software (Nikon Corporation). The FRET ratio for fluorescence emissions of CFP and YFP ($F_{YFP}/F_{CFP}$) for single cells was calculated and corrected for background, bleed-through, and photobleaching as previously described[46]. Individual cells were perfused with buffer or with the ligand for the time indicated by the horizontal bar shown in plots.

## Membrane preparation and PIP$_3$ pull down

The principle of the experiment is depicted in Fig. 7A. Membrane and cytosolic fractions were prepared as previously described[6]. HEK293 cells expressing HA-tagged PTH$_1$R were seeded on poly-D-lysine coated 10 cm plates a day before the experiments were performed. Cells were incubated in serum-free DMEM for 1 h and then stimulated with 100 nM PTH$_{1-34}$ in DMEM for 5 min or left untreated. Cells were washed three times in cold PBS, scrapped, and transferred to a microcentrifuge tube. Cells were centrifuged at 200 x g for 2 min; the supernatant was discarded, and the pellet was resuspended in PBS supplemented with protease inhibitors. Cells were lysed by sonication in short pulses of 15 sec 4 times in a bath Bioruptor Pico sonicator (Diagenode). Cell lysates were centrifuged at 1000 x g at 4 °C for 10 min to remove unbroken cells. The supernatant was then centrifuged at 20,000 x g at 4 °C for 30 min. The subsequent pellet corresponds to the membrane fraction, while the supernatant is the cytosolic fraction. Fractions were incubated overnight at 4 °C with PIP$_3$-coated or control beads (Echelon Bioscience catalog# P-B003A). Beads were washed, resuspended in 2X sample buffer with β-mercaptoethanol, and loaded onto an SDS-PAGE gradient gel.

## PIP$_2$ and PIP$_3$ measurements

HEK293 cells were transfected with 0.5 μg of PTH$_1$R[CFP] and mCherry-aPHx2 for 24 h. Time-lapse recordings were acquired by 12 min on a TIRF microscopy. Before PTH stimulation, 30 s were used as a basal level. Background subtraction was performed in all images before quantification. Regions of interest (ROIs) were generated covering the cells and the fluorescence change in mCherry-aPHx2 sensor was monitored. The fluorescence intensity was normalized using the average fluorescence of the first 30 s before stimulation with PTH.

## PTH$_1$R cluster formation in CCP

HEK293 cells stably expressing PTH$_1$R-GFP were transfected with 0.5 μg CLC-DsRed for 24 h. Time-lapse recordings were acquired on a TIRF microscope for 35 min. Before PTH stimulation, 10 s were used as a basal level. The background was subtracted and PTH$_1$R clusters were segmented using the "Find Maxima" algorithm, Image J, on the image acquired 5 s post PTH challenge and saved as ROIs. PTH$_1$R-GFP and CLC-DsRed intensity were quantified within each ROI. Fluorescence levels were normalized by the initial intensity recorded in each ROI.

## Dynamics of βarr1 recruitment to PTH$_1$R clusters

HEK293 cells stably expressing PTH$_1$R-GFP were transfected with 0.5 μg βarr1-Tomato for 24 h. Time-lapse recordings were acquired on a TIRF microscope for 125 s. Before PTH stimulation, 10 s were used as basal level. The background was subtracted and PTH$_1$R clusters were segmented using the "Find Maxima" algorithm, Image J, on the image acquired 5 s post-PTH challenge and saved as ROIs. PTH$_1$R-GFP and βarr1-Tomato intensities were quantified within each ROI. Fluorescence levels were normalized by the initial intensity recorded in each ROI.

## β-arrestin expression and purification

The parent Cys-less human β-arrestin 2 (βarr2) (C17S, C60V, C126S, C141L, C151V, C243V, C252V, C270S, C409S) with a single cysteine reintroduced in the finger loop (L69C) was prepared as previously described[25], as was the the variant βarr2 L69C/K233Q/R237Q/K251Q.

Briefly, this construct is modified with an N-terminal 6x Histidine tag, followed by a 3 C protease site, a GG linker, AviTag and GGSGGS linker. The sequence was codon-optimized for expression in *E. coli* and cloned into a pET-15b vector. βarr2 L69C was expressed as follows: NiCo21(DE3) competent *E. coli* (NEB) were transformed, and large-scale cultures were grown in TB + ampicillin at 37 °C until an $OD_{600}$ of 1.0. Cells were then transferred to room temperature and induced with 25 μM IPTG when the $OD_{600}$ reached 2.0. Cells were harvested 20 h post-induction and resuspended in lysis buffer [50 mM HEPES pH 7.4, 500 mM NaCl, 15% glycerol, 7.13 mM 2-mercaptoethanol (BME)][47] to a final volume of 40 mL/L of cells. Cells were lysed by sonication and the clarified lysate was applied to nickel sepharose and batch incubated for 1.5 h at 4 °C. The resin was washed with 10 column volumes of wash buffer (20 mM HEPES pH 7.4, 500 mM NaCl, 10% glycerol, 7.13 mM BME) + 20 mM imidazole, followed by 10 column volumes of wash buffer + 40 mM imidazole. The protein was then eluted with 5 column volumes of wash buffer + 200 mM imidazole and dialyzed overnight in 100x volume of dialysis buffer (20 mM HEPES 7.4, 200 mM NaCl, 2 mM BME, 10% glycerol) in the presence of 1:20 (w:w) of 3 C protease. The digested protein was then subjected to reverse-Nickel purification and diluted with dialysis buffer containing no NaCl to bring the NaCl concentration to 75 mM. The protein was then purified by ion exchange chromatography (mono Q 10/100 GL, GE Healthcare), followed by SEC using a Superdex 200 increase 10/300 GL column (GE Healthcare) with SEC buffer (20 mM HEPES pH 7.4, 300 mM NaCl, 10% glycerol). Purified protein was concentrated to between 100–300 μM using a 30 kDa spin concentrator and aliquots were flash-frozen in liquid nitrogen and stored at −80 °C until use.

### β-arrestin labeling

Following SEC, elution peak fractions were pooled to a concentration of 10–20 μM and labeled with monobromobimane (mBBr) as follows. mBBr was dissolved in DMSO to 25–40 mM and added at 10x molar excess over protein, then allowed to react for 1 h at room temperature before quenching with L-Cysteine (10x molar excess over fluorophore). The labeling reaction was further incubated for 10 min after cysteine addition, after which samples were spin-filtered and subjected to a second round of size-exclusion chromatography, as detailed above, to remove the free dye. The purified protein was concentrated to between 100–300 μM using a 30 kDa spin concentrator and aliquots were flash-frozen in liquid nitrogen and stored at −80 °C until use.

### Bulk fluorescence measurements

Bulk fluorescence measurements were performed using either a Tecan Infinite M1000 PRO or a BMG CLARIOstar Plus multimodal microplate reader. Equilibrium spectra were collected by mixing the corresponding βarr2 proteins with buffer or ligand (diC8-PI(3,4,5)P3, di-C8-PI(4,5)P2, V2Rpp) and incubating for 1 h in the dark before measurement. Bimane spectra were collected using 96-well ½ area white plates with 50 μL of sample and at a final concentration of 0.5 μM βarr2 in buffer containing 20 mM HEPES pH 7.4, 100 mM NaCl, and 0.004% LMNG/0.0004% CHS (w/v). For the M1000 PRO, the following instrument settings were used: excitation: 370 nm, emission 420–500 nm (2 nm steps) with 200 reads and 500 μs read time in 400 Hz flash mode. Gain and z-position were optimized before reading. For the CLARIOstar Plus, the following instrument settings were used: excitation: 370 nm, emission 420–500 nm (10 nm slit width for both excitation and emission), 2 nm scan increments, 0.5 s settle time, and 20 flashes per read. Gain and z-position were optimized before reading for each experiment. ΔF/F vs [PIP$_3$] (or [PIP$_2$]) was fit using GraphPad Prism to a single site binding model with $\Delta F/F = F_{max} \times [PIP_3]/(K_d + [PIP_3])$ where $F_{max}$ is the maximum F, [PIP$_3$] is PIP$_3$ concentration, and $K_d$ the PIP$_3$ concentration needed to achieve 50% of $F_{max}$.

### Photometric recording of time courses of PTH[TMR] binding in single-cell assays

Time courses of TMR-labeled peptide binding to mock HEK293 cells were recorded using an emission photometric system previously described[46]. Cells plated on poly-D-lysine-coated glass coverslips maintained in HEPES/BSA buffer (HEPES buffer containing 0.1% (w/v) BSA) were placed at room temperature on a Zeiss inverted microscope (Axiovert135) equipped with an oil immersion 100X Plan-Neofluar objective and a dual emission photometric system (TILL Photonics, Planegg, Germany). Cells were excited with light from a polychrome V (TILL Photonics). A single cell was continuously superfused with HEPES/BSA buffer without or with PTH[TMR] using a computer-assisted solenoid valve rapid superfusion device (ALA-VM8, ALA Scientific Instruments, solution exchange 5 to 10 ms). Signals detected by avalanche photodiodes were digitalized using an analog/digital (AD) converter (Digidata1433A, Axon Instruments) and stored on a personal computer using Clampex 11.0 software (Axon Instruments). The illumination time was typically set to 25 ms applied with a 5 Hz frequency. PTH[TMR] fluorescence was recorded at 580 ± 20 nm upon excitation at 550 nm (D540/25x, beam splitter 555 DCLP, emission filter HQ605/75).

### Statistics and reproductibility

Statistical analyses were processed using GraphPad Prism version 9 (GraphPad Software, La Jolla California USA). Data were expressed as mean ± SD, 95% confidence intervals, or SEM. Unless indicated, the mean per cell is graphed to produce a grand mean. For diffusion coefficients calculated from MSD, the grand mean represents the average of the median diffusion coefficients per cell. No statistical method was used to predetermine sample size. The Investigators were not blinded to allocation during experiments and outcome assessment. Statistical analyses for data involving one independent variable and three groups were performed using One-way ANOVA with Tukey's or Dunnett's multiple comparison posthoc tests. For two groups of data, a two-tailed *t*-test was used.

### Reporting summary

Further information on research design is available in the Nature Portfolio Reporting Summary linked to this article.

## Data availability

The source data that support the results from the current study are stored in files created with Excel 2013 (Microsoft Corp., Redmond, WA) and GraphPad Prism version 9 (GraphPad software, La Jolla, CA). Source data are provided with this paper.

## Code availability

Codes employed for the analysis are available on Github [https://github.com/JPacheco5/Single-molecule-tracking].

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

## Acknowledgements

Research reported in this publication was supported by the National Institute of Diabetes and Digestive and Kidney Disease (NIDDK), the

National Institute of General Medical Sciences (NIGMS), and the National Institute of Dental and Craniofacial Research (NIDCR) of the US National Institutes of Health (NIH) under grant Award Numbers R01-DK116780 (to J.-P.V.), R01-DK122259 (to J.-P.V.), R21-DE032478 (to J.-P.V.), R35-GM119412 (to G.R.V.H.), K99-GM147609 (to J.J.) and T32-GM133332 (to S.S.). The authors thank Thomas Gardella and Ashok Khatri (Massachusetts General Hospital and Harvard Medical School) for providing TMR-labeled peptides, and Elizabeth White for assistance with βarr2 expression and purification.

## Author contributions

Conceptualization: J.P.V., J.P., K.A.P.; Methodology: J.P., K.A.P., G.R.V.H., J.J., J.P.V.; Investigation and Data analysis: J.P., K.A.P., S.S., A.G., J.J., J.P.V.; Project administration: J.P.V.; Supervision: J.P.V.; Writing—original draft: J.P.V.; Writing—review and editing: K.A.P., J.P., J.J., J.P.V.

## Competing interests

The authors declare no competing interests.
