## [Transparent Peer Review file · Nature Communications]

Fast-diffusing receptor collision with slow-diffusing peptide ligand assemble the ternary parathyroid hormone–GPCR–arrestin complex

Corresponding Author: Professor Jean-Pierre Vilardaga

Version 1:

Reviewer comments:

Reviewer #3

(Remarks to the Author)

Pacheco et al have provided a revised version of their manuscript, which addressed some, but not all of the concerns that were addressed in a previous revision.

Some of the new experiments that appear to better support their claim, are unfortunately missing some important controls. Since the suggestion in previous rounds of review had been quite specific, I take the liberty of reiterating the specific experiments that would allow to allay any remaining concern:

1. In the 'flipped geometry' the top membrane becomes again a bottom membrane, i.e. it is sandwiched and pushed against the coverslip, so any concern related to the original bottom membrane, would also apply here, such as geometrical ligand confinement. I therefore suggest to remove the connotation that these experiments report on the cell apical membrane.
2. A very specific experiment had been suggested, i.e. the use of a different labeled 'mock' peptide to address the issue of the potential non specific binding. Ideally a peptide other than PTH, lacking the sequence BXXXXXXB should be imaged by single particle tracking .
3. Use of diffusion to discriminate ligand bound/stuck to surface from ligand embedded in the membrane is not conclusive, since the ligand diffusing in the gap between the cell membrane and the coverslips (either bottom or top) could have an additional mode of motion.
4. On the other hand, the confocal FRAP in Figure 11-J looks promising. Can we have FRAP data also on the receptor-neonG channel, please?
5. The explanation of the mechanism hinges on Figure S6, which, as discussed below, raised some issues of consistency with other data presented in the manuscript. Crucially, it would be now important to conduct FRAP experiments on cells treated with hyaluronidase.
6. Figure S6A dissociation of the ligand: in native conditions, most ligand is dissociated within 2 minutes. However, there is recovery at 200 s (3 minutes) shown in Figure 11. How do you interpret FRAP data and mobile fractions of a rapidly dissociating ligand?

(Remarks on code availability)

Reviewer #4

(Remarks to the Author)

Pacheco, Vilardaga, and colleagues present an interesting study in which they use single molecule TIRF imaging—with supporting biochemistry—to argue against the classic text-book model of GPCR activation. Instead of the ligand diffusing

through solution to the receptor, the authors show that fluorophore-labeled PTH associates with the cell membrane and the encounter with PTH1R is through diffusion of the receptor to the relatively immobile ligand. Again, unlike the text-book model, the authors argue that the primary association of the receptor-ligand complex with B-arrestin is at the CCP (rather than association of the tripartite complex prior to CCP entry). The study is impressive (the inverted TIRF setup for imaging the top of the cell is particularly clever) and extensive controls have been added during previous rounds of review. Of particular note are the line of new control experiments depleting hyaluronic acid and showing reduced TMR-PTH association with the cell membrane (it would be nice if these data weren't buried in supplement, as they are an important control).

I think this is an excellent study and I would recommend it for publication after addressing my few relatively minor criticisms.

1) Figure 4. I am surprised that formation of the B-arrestin/PTH1R complex is fully blocked with TGX-221 treatment (Figure 4C). The implication from Figure 4 is that agonist-induced endocytosis of PTH1R would also be fully blocked by TGX-211? I thought the important control experiment (PTH1R endocytosis assays using TGX-211) might be present in this manuscript or White et al., 2020, but I couldn't find it. Can the authors address this concern? Ideally, with flow-cytometry based assays for agonist-induced PTH1R endocytosis with and without TGX-211. My concern is that—as the authors know—many non-Gq coupled receptors recruit B-arrestin (presumably without the PIP3 mechanism described here and in White et al. 2020). I would have assumed that those mechanisms (Gbeta/gamma with GRK2/3, C-edge, etc) could still come into play here even if PIP3 levels aren't increased and that these mechanisms would still allow for PTH1R/B-arrestin association and subsequent endocytosis (albeit, at lower levels than with Gq/PIP3). If the authors find that agonist-induced endocytosis of PTH1R isn't fully blocked by TGX-211 treatment, the implication is that there are other—biologically relevant—PTH1R/B-arrestin interactions the authors aren't capturing in their single molecule assays, which would be important to note in the text.

2) Better comparison with Grimes et al., Cell, 2023. One of the two critical findings of this paper is that B-arrestin is recruited to the PM and CCP (largely) independently of receptor-ligand (RL) complex. A similar finding was recently shown (Grimes et al., 2023) for B2AR. Considering the broad similarity of these studies and techniques, the authors discussion of Grimes—relative to their own paper—is important. However, I found the discussion about Grimes quite confusing. Please re-write/clarify this section of the discussion to underscore the similarities between the studies (B-arrestin is a largely independent of the receptor in its movement to CCPs) and highlight the differences regarding B-arrestin (C-edge vs PIP3, etc). Also, I found it surprising that Eichel et al., 2016 and 2018 weren't cited considering these studies showed early evidence for B-arrestin acting independently of RL at the CCP.

3) Explanation of PIP-resin experiments. I find part of the PIP resin experiments confusing (Figure 5B). Can the authors offer a rational/information for: A) Why activation with PTH added before lysis (and before at 18 hour pulldown) causes a persistent change in the ability of B-arrestin to binding PIP3 resin (lanes 3 and 4 much higher than lane 1 or 2 as seen Figure 5B and 5C). Do the authors think B-arrestin is activated through a PTM, and that is the persistent activating factor? I would assume any cell biological changes would likely have been long lost during the incubation with resin (and even if it wasn't not, changes like increased cellular PIP3 should have acted as a competitive inhibitor to binding PIP3 resin). B) Why is the total amount of PTHR-HA (input blot) greater in the PM and C fractions when agonist is added (lanes 3 and 4) compared to no agonist (lanes 1 and 2)? My concern here is that the data suggest that adding agonist improves the ability of B-arrestin to bind PIP3 resin (as shown in quantification in 5C, grey bars vs red); however, if there is a different amount of PTH1R in the 10cm plates, direct comparison of agonist and no-agonist could be misleading; C) I couldn't find information in the material and methods on where the PIP resin was purchased (I assume Echelon Biosciences?).

4) Regarding previous rounds of review: The idea that PTH associates with the cell, and then encounters PTH1R through lateral diffusion of the receptor, is surprising. The authors have added many controls from the previous rounds of review to strengthen their argument, including: 1) the choice of fluorophore (TMR and fluorescein) doesn't alter PTH foci; 2) addition of unlabeled PTH also causes slowing of PTH1R; 3) a mechanistic explanation with hyaluronic acid manipulation. I am generally satisfied and do not think additional controls are required.

(Remarks on code availability)

Reviewing the code is outside my field of expertise.

Version 2:

Reviewer comments:

Reviewer #3

(Remarks to the Author)

please see enclosed PDF file

(Remarks on code availability)

Reviewer #4

(Remarks to the Author)

These revisions address all my concerns, and the current manuscript is acceptable.

For the authors benefit, I noticed two typos in the newly added text

Line 384: "sequential steps of Barr ____ before"
Line 394 "before to collide"

(Remarks on code availability)

We express our gratitude to the reviewers for their feedback on our manuscript and helpful suggestions for its enhancement. We have made significant revisions to the manuscript and figures and have included new data to take into account the feedback and provide more clarity where necessary. Please note that the major additions in the revised manuscript are in blue.

Responses to general comments from reviewers

Slow mobility of PTH^{TMR}

The reviewers are concerned that the immobility comes from PTH binding to the glass coverslip, however, we observed the same slow mobility of PTH-TMR on the top surface of the cell, not in contact with the coverslip (**Fig. 1D, E**).

Through a detailed diffusion analysis, we were able to identify significant differences in the diffusion properties of PTH^{TMR} molecules that were bound to the coverslip and the cell membrane. The spatial control allowed us to determine that PTH^{TMR}, which was bound to the fibronectin-coated coverglass, exhibited slower diffusion in comparison to the ligand that was bound to the cell membrane.

To ensure that our results are not influenced by PTH^{TMR} attached to the coverslip, we applied three levels of stringency. First, we took measurements restricted to the cell membrane. Secondly, we focused on PTH^{TMR} molecules that colocalized with the PTHR^{mNG}. Finally, we excluded from our analysis all fluorescent spots with diffusion coefficient values that fall below the threshold value corresponding to PTH^{TMR} non-specifically attached to the fibronectin-coated glass coverslip. We have revised the text on pages 4 and 5 as follows:

“To differentiate PTH^{TMR} molecules bound to the cover glass from those in cells, we analyzed the diffusion properties of the ligand both inside and outside the cell (**Fig. 1C** and **S5A-C**). We analyzed the diffusion properties of the ligand both inside and outside the cell (**Fig. 1C** and **S5A-C**). The diffusion coefficient values of PTH^{TMR} were similar in cells expressing PTH_{1R} cells and in mock cells ($D = 0.0018 \pm 0.0006$ and $0.0015 \pm 0.0001 \mu\text{m}^2/\text{s}$, respectively), but significantly ($P = 0.0008$) distinct from the value of PTH^{TMR} non-specifically attached to the glass coverslip ($D = 0.00059 \pm 0.00011 \mu\text{m}^2/\text{s}$) and which are considered immobile (**Fig. 1C**). We thus used a threshold in our analysis to differentiate PTH^{TMR} on coverslips from that on cells. We excluded all fluorescent spots with diffusion coefficient values below the threshold diffusion $D = 0.00059 \mu\text{m}^2/\text{s}$ (horizontal dotted line in **Fig. 1C**) plus two standard deviations. The diffusion coefficient of all filtered PTH^{TMR} trajectories was similar to the diffusion of trajectories that colocalized with PTH_{1R}^{mNG}, confirming the accuracy of our analysis (**Fig. S5D, E**).”

Mechanism

Reviewers requested a mechanistic explanation of our results. Our most recent findings, as shown in Figure S6, suggest that the attachment of PTH to the plasma membrane depends on hyaluronic acid (HA), a glycosaminoglycan (GAG) present on the cell surface. When mock cells were treated with hyaluronidase (HAase), they exhibited significantly less PTH^{TMR} binding compared to untreated cells. However, other factors may play a role in retaining PTH on the cell surface, since PTH binding was not completely blocked by HAase treatment. Based on these results, we think that the ligand-bound receptor is formed after receptor collision with the low mobile HA-bound PTH ligand. Additionally, Fig. 7 reports our analysis of β arr2 diffusion into clathrin spots to determine β arr2 diffuses independently of the PTH-bound PTH_{1R}. This is described on pages 13/14 and discussed on pages 15/16 as follows:

(Results p13/14) “To further investigate the remaining fraction of β arr2 that is not associated directly with CCPs (~30%), we asked whether PTH₁R, β arr2, and PTH can diffuse into pre-existing clathrin clusters separately or together in a complex. To address this point, we used a geometric approach. First, we used diffusion coefficient and motion values of PTH₁R^{mNG} and β arr2^{mNG} in basal condition (i.e., no PTH), and in complex (with PTH) using our dual-color interacting trajectories of either β arr2^{mNG} and PTH^{TMR} in cells expressing PTH₁R, or β arr2^{mNG} and PTH₁R^{IRFP} in the presence of PTH (**Fig. 7A** and **Table S3**). Then, we generated binary masks from the distribution of clathrin spots to identify areas without clathrin clusters. Using this information, we constructed a Voronoi diagram to determine the radius of all empty circles (**Fig. 7B–D**). Finally, we generated a plot reporting MSD vs time based on the calculated diffusion and anomalous motion values of PTH₁R and β arr2 freed of PTH, and interacting PTH₁R, PTH, and β arr2. The results indicate that PTH₁R and β arr2 molecules explore an area with a velocity of 6.9 $\mu\text{m}^2/\text{min}$ and 0.4 $\mu\text{m}^2/\text{min}$, respectively, in the absence of PTH. If located in the center of an averaged empty clathrin surface of $\sim 0.4 \mu\text{m}^2$, PTH₁R and β arr2 can reach the border of a clathrin domain in ~ 1.1 and 56 seconds, respectively (**Fig. 7E**). However, the area of exploration of the PTH-PTH₁R- β arr2 complex dropped down to 0.03-0.06 $\mu\text{m}^2/\text{min}$ suggesting that the complex can only reach nearby clathrin clusters (**Fig. 7E**). These results suggest that the PTH-PTH₁R- β arr2 complex is more likely to assemble within clathrin clusters (**Fig. S13**).”

(Discussion p15/16) “The assembly involves a sequence of steps and molecular interactions at the plasma membrane that starts when PTH₁R collides and binds PTH via lateral diffusion. This collision can take place either outside or inside clathrin spots given that PTH is slowly diffusing and randomly distributed with $\approx 50\%$ of molecules encountered in clathrin clusters. The PTH-bound receptor induces activation of Gq and PI3K β resulting in the production of PIP₃, which in turn stabilizes the β arr2 contact with the PTH₁R. Biochemical results reported in **Fig. 5** and the time delay before β arr2 and PTH₁R collisions observed in **Fig. 6**, suggest PIP₃ may promote conformational rearrangements in β arr2 before its interaction with the PTH-bound receptor. Then, β arr2 collides with the PTH-bound receptor through lateral diffusion. A recent study further supports the association of β arr2 with the plasma membrane before its interaction with the β_2 -adrenergic receptor (β_2 AR) in clathrin-coated pits. However, β arr2 initially binds to the plasma membrane, undergoes lateral diffusion, and engages its active conformation through random collisions with active receptors. Consequently, the dwell time of β arr2 at the plasma membrane increases after accumulation of β arr2 in CCPs where it associates with the β_2 AR. This model postulates that β arr2 and receptor diffuse to CCPs separately (31). In the case of the PTH₁R, the high fraction ($\approx 70\%$) of β arr2 molecules located inside clathrin clusters when PIP₃ is elevated (**Fig. 6**) suggests that most of the PTH-PTH₁R- β arr complex is located inside clathrin-coated pits. Based on the predictive kinetics of β arr2 diffusing into CCPs shown in **Fig. 7**, the accumulation of the PTH-PTH₁R- β arr complex in clathrin clusters is slower than β arr2 diffusing into pre-existing clathrin clusters separately from the PTH-PTH₁R complex. In conclusion, our findings suggest that the ligand-bound receptor is formed after receptor collision with the slow diffusing peptide ligand. Our results further support a model where β arr2 recruitment to the plasma membrane and its interaction with PTH₁R requires PIP₃ and clathrin clustering.”

Reviewers Comments:

Referee #1 (Remarks to the Author):

The authors have done a fine job in addressing most of my comments. I feel that the additional controls and experimental data sets have substantially improved the quality of the present manuscript. However, I still have several comments to the revised version of the manuscript that need to be addressed.

1) The finding that the PTH peptide is trapped in the cell membrane and, thus, essentially immobile is a very striking and unprecedented result. The new data provided by the authors do indeed strengthen the initial observation. However, I am not entirely convinced by the single-molecule images shown in Supplementary Figure 3. Here, the authors show that PTH-TMR addition to uncoated (row 2) and fibronectin-coated (row 3) coverslips results in high fluorescence background that is, visually, indistinguishable from PTH-TMR addition to HEK-PTH1R cells (row 5). The authors claim that the PTH-TMR on coverslips (no cells) is even more immobile than the fraction they observe upon addition to cells. However, it is not clear, given this dramatic unspecific background, how the authors know which fluorescent spot is a ligand-bound receptor and which spot is unspecific background. I am concerned that the immobility of PTH-TMR in samples with HEK cells expressing PTHR1 (the 'real' experiment) is overestimated or even misinterpreted due to the presence of this high immobile and unspecific PTH-TMR background.

Response: see our general response.

2) If the mechanism of PTH-TMR immobility was correct, it would suggest that cells can use ligand-trapping to localize active receptors only to certain hot-spots. As PTH receptors couple to b-arrestin in those hot-spots, this mechanism would be a way to ensure internalization and to foster subsequent PTHR-signaling from endosomes. Given this interesting mechanism, it would be important to provide at least some data on the generalizability of these findings: is this mechanism conserved in all peptide receptors, or only in ClassB receptors, or only those that are prone to endosomal signaling?

Response: This is a limitation of our study that focuses on the PTH1R noted at the end of the discussion.

The manuscript lacks any sound mechanistic explanation which I feel is required for a truly convincing story.

Response: see our general response.

3) If the proposed mechanism is correct, where and how would PTHR couple to G proteins? Would this also occur in PTH-containing, immobile CCPs? The manuscript would profit from a discussion on how canonical G-protein coupling would fit into the proposed mechanism.

Response: The PTH1R collides and binds PTH through lateral diffusion. This collision can occur both inside and outside clathrin clusters, as PTH diffuses slowly and is randomly distributed with 50% of molecules found in clathrin spots. We anticipate that the PTH-bound receptor induces activation of Gs and Gq located inside and outside of the clathrin-coated pit. The PTH-bound receptor induces activation of Gq and PI3K β resulting in the production of PIP₃, which in turn stabilizes the β arr2 contact with the PTH₁R. Biochemical results reported in Fig. 5 and the time delay before β arr2 and PTH₁R collisions observed in Fig. 6, suggest PIP₃ may promote conformational rearrangements in β arr2 before its interaction with the PTH-bound receptor. Then, β arr2 collides with the PTH-bound receptor through lateral diffusion. A recent study further

supports the association of β arr2 with the plasma membrane before its interaction with the β -adrenergic receptor (β ₂AR) in clathrin-coated pits. However, β arr2 initially binds to the plasma membrane, undergoes lateral diffusion, and engages its active conformation through random collisions with active receptors. Consequently, the dwell time of β arr2 at the plasma membrane increases after accumulation of β arr2 in CCPs where it associates with the β ₂AR. This model postulates that β arr2 and receptor diffuse to CCPs separately. In the case of the PTH₁R, the high fraction (\approx 70%) of β arr2 molecules located inside clathrin clusters when PIP₃ is elevated (Fig. 6) suggests that most of the PTH-PTH₁R- β arr complex is located inside clathrin-coated pits. Based on the predictive kinetics of β arr2 diffusing into CCPs shown in Fig. 7, the accumulation of the PTH-PTH₁R- β arr complex in clathrin clusters is slower than β arr2 diffusing into pre-existing clathrin clusters separately from the PTH-PTH₁R complex. In conclusion, our findings suggest that the ligand-bound receptor is formed after receptor collision with the slow diffusing peptide ligand.

Referee #2 (Remarks to the Author):

Overall the authors did an excellent job of addressing the concerns that I had and the manuscript is clearer and has more critical controls and mechanistic considerations. I have only a couple of minor comments.

1) The language in this section is unclear, although I think I know what they mean. From a logic standpoint if PIP3 is like PIP2 and V2Rpp why would it be more effective? You mean because it has a higher affinity for PIP3 than PIP2? I assume.

“Like PIP2 and V2Rpp, PIP3 induced a concentration dependent increase in bimane fluorescence suggestive of a conformational change of the β arr2 consistent with activation (Fig. 5D, E). This suggests that PIP3 could be more effective for increasing and retaining the pool of β arr2 at the plasma membrane than PIP2 (Fig. 5F).”

Response. Yes, this is correct. This point has been added on page 12.

2) I find this paragraph confusing and it should be rewritten. How can ligand bound receptor not be formed through direct ligand receptor interactions? The last sentences are also not clear.. What part of all of this is proposed as an alternate to the central tenet? this could use some grammatical clarification.

Response. We have revised our discussion to avoid any confusion.

Referee #3 (Remarks to the Author):

In the revised version of the manuscript PTH-GPCR-arresting complexes assembly in selective cell membrane spots, Pacheco et al have addressed in part the comments raised by the three reviewers.

I will focus here preliminarily on the key controls that were required from my side, namely to rule out ligand (PTH-TMR) interaction with the surface (coverslip) as the source of the observed reduction in diffusion dynamics.

As far as I am concerned, the rebuttal to this concern is presented by the authors in Extended Data Figure 2/3 and in Supplementary Figure 3 (which should not be tucked away in the supplementary, but rather be an Extended Data Figure itself).

Response: Controls are shown in Figs. 1, S4, and S5.

Supplementary Figure 3: as such, the Figure still lacks some key information for its proper interpretation. Spatial scale is missing (what is the size of the scale bar, and is it the same for all images)? What is the timescale of the measurements, i.e. at what time are the subsequent snapshots taken?

Response: Fig. 1 (previously Fig. S3) addresses the spatial scale, timescale, grayscale, and time point of ligand incorporation.

When there are cells on the coverslip, where are they (i.e. Transmitted light or DIC is missing, and is critical in assessing the amount of aspecific spots below vs outside the cells)

Response: To achieve the correct focus in the z-axis under TIRF illumination, we performed the detection of PTH^{TMR} with respect to the focus plane of PTHR^{mNG} or in cells that were endogenously labeled with CLTA^{mNG}. This ensured that the excitation plane was correctly aligned with the single molecule detection. We have updated the figures to include both channels.

Moreover, are these representative images? If so, how many experiments? It would be important to have statistics of the number of spots measured / unit area for all conditions.

Response: Figure S4 addresses reviewer's point and shows density measurements to provide a robust comparison of the PTH^{TMR} binding.

Besides, these crucial technical aspects, the first question I would like the authors to clarify is why TMR only was presented in one condition (PTHR cells on Fibronectin)-

Response: the TMR control was shown in the conditions where experiments were performed.

Next puzzling issue is the observation of PHTMR saturating the images when added to uncoated coverslips (no cells). This would indicate a very strong aspecific binding of the PTH-TMR to the surface. 'Fibronectin (no cells)' does not display this major effect, but still there is sizable non-specific binding, i.e. immobile spots, which is overall the most concerning outcome of this control.

Their spatial density, based only on these few representative snapshots, is larger than that observed in 'mock cells', and appears of the order 30-50% of the number of spots observed underneath PTHR-cells on fibronectin coated coverslips. This means that in any single particle analysis, about half of the immobilized spots observed are due to aspecific surface binding, and thus that a large proportion of the immobile traces observed in the experiments reported in the main manuscript are due to immobile PTH-TMR at the surface.

Response: see our general response.

If the model is that PTH ligand binds to the cell membrane prior to encounter with the ligand, the comparison of PHTMR binding to Mock cells vs PTHR cells is puzzling, since binding to mock cells appears delayed, suggesting that the predominant route for PTHR binding is indeed direct receptor binding. Again, any further speculation on this aspect would require a clear indication of a temporal scale, outlines of the cells, as well as statistics on the number of spots and more representative movies.

Response: Figure S4 addresses this point. Here, we conducted a detailed analysis of temporal courses, quantifying the density of molecules. Our results indicate that there is no delay in PTH^{TMR} binding to mock cells. While mock cells indeed bind fewer PTH molecules compared to cells expressing the receptor, the diffusion measurements demonstrate that PTH^{TMR} behaves similarly on the membrane of both mock and PTHR-expressing cells.

Extended Data Figure 2: FRAP experiments at the membrane of the cells indicate an approximately 20% recovery of the ligand signal after bleaching, suggesting that approximately 80% of the ligand is bound. These data need some controls, that are unfortunately missing. The easiest one, prominently absent, is the recovery of the mNg tagged receptors (without PTH-TMR): what extend of recovery do the receptors show? It is unlikely to be 100%, in light of the considerations about scaffolding that the authors raise in their rebuttal point 1.9.

Response: This is correct. We did not observe 100% recovery of receptor fluorescence in the absence of PTH. However, the recovery of fluorescence was reduced by PTH, corroborating our single-molecule diffusion measurements in Figure 2.

Additionally, it would be cool to observe control recoveries for previously used receptors (e.g. Dorsch et al 2010), such as CD86, or the beta2-AR.

Response: A limitation of our study is to focus on PTH1R. As stated at the end of our discussion, further research is needed to determine if the mechanism reported for the PTH1R can be applied to other peptide ligand GPCRs.

There are several minor aspects

1.8 The answer to this legitimate point from reviewer 1 is very technical. Nonetheless, the reviewer is right in wondering why the supposedly immobilized receptor+ ligand goes faster than the immobilized ligand alone

Response: We propose that hyaluronic acid is a possible extracellular component in facilitating PTH retention at the cell surface (Fig. S6). This HA-dependent PTH binding to the cell surface is likely responsible for the slow mobility of the ligand. We assume that PTH dissociates from HA after its collision with the receptor resulting in a higher PTH mobility when bound to PTH1R.

3.1.1: By using other peptides, I intended peptides radically different from PTH.

3.1.2 Other peptidic GPCR ligands were not tried

Response: This can be considered a limitation of our study. Further experiments are needed to demonstrate whether the immobile-PTH feature also applies to other peptide ligand-receptor systems.

3.1.3 Re FRAP, see my general comments above

Response: see our response above.

3.2: Albeit both myself and reviewer #1 asked for a mechanism for the PTH immobilization, the

authors are adamant that they are not providing one.

Response: see our response to general comments.

3.3: Cool snapshots, but what about whole-cell movies, showing labelling efficiency, density, nonspecific coverslip binding...

Response: We have addressed this point by adding movie 1.

3.6 The justification of why intensity values (the authors do not measure brightnesses) for dimers do not double with respect to the corrected monomeric values is not entirely clear to me. There are well established methods to measure this, and most dimerization-related SMT works can show controls where dimers have double the intensity of the monomers. As an afterthought, did the authors remember to subtract background counts?

Response: The absence of an arithmetic increase in fluorescence could also be explained by the relationship between the signal-to-noise ratio (SNR) and photon counts in our CMOS sensor. This relationship is not linear, as the SNR generally increases with higher photon counts. However, under conditions of low fluorescence intensities, the SNR decreases, impacting the observation of a perfect arithmetic increment in fluorescence. For our analysis, a background subtraction was performed by using a wavelet filter. Note that we have removed the part describing PTH1R stoichiometry (previous Fig. 2) that we think is not necessary for this manuscript.

3.10 The question was about temporal gaps, and is not addressed.

Response: This information was added to the new version of the manuscript (Page 18): "We used 0.5 \$\mu\text{m}\$ for linking and gap-closing with an implementation of a simple linear assignment problem algorithm".

We are grateful for the thoughtful comments provided by the reviewers. We have taken these comments into careful consideration and ensured that all concerns have been thoughtfully addressed.

Reviewer #3

Pacheco et al have provided a revised version of their manuscript, which addressed some, but not all of the concerns that were addressed in a previous revision. Some of the new experiments that appear to better support their claim, are unfortunately missing some important controls. Since the suggestion in previous rounds of review had been quite specific, I take the liberty of reiterating the specific experiments that would allow to allay any remaining concern:

1. In the 'flipped geometry' the top membrane becomes again a bottom membrane, i.e. it is sandwiched and pushed against the coverslip, so any concern related to the original bottom membrane, would also apply here, such as geometrical ligand confinement. I therefore suggest to remove the connotation that these experiments report on the cell apical membrane.

Response. We greatly appreciate your comment, which allows us to include some information in the methodology. Our approach involved adding the fluorescent ligand to the cells first, then removing it (wash off). We then inverted the coverslip to implement the TIRF configuration. This method ensures the ligand's attachment to the membrane, preventing any residual ligand from floating. The revisited manuscript now provides a more detailed account of our experiment procedure on page 5

2. A very specific experiment had been suggested, i.e. the use of a different labeled 'mock' peptide to address the issue of the potential non specific binding. Ideally a peptide other than PTH, lacking the sequence BXXXXXB should be imaged by single particle tracking .

Response. The conclusions put forward by the reviewer may not be directly relevant to PTH behavior. We are seeking further understanding of whether the experiment involves randomization of the PTH sequence or another peptide unrelated to PTHR. Our findings indicate that PTH₁₋₁₄, which does not contain a potential BXXXXXB motif, exhibits a weak attachment to the cell surface compared to PTH.

3. Use of diffusion to discriminate ligand bound/stuck to surface from ligand embedded in the membrane is not conclusive, since the ligand diffusing in the gap between the cell membrane and the coverslips (either bottom or top) could have an additional mode of motion.

Response. We understand that there may be some amount of ligand present in the aqueous phase. However, in our particular case, this amount is not significant because several studies have shown that diffusion in aqueous phases is $>10 \mu\text{m}^2/\text{s}$ (1), which is much faster than our detection speed. Therefore, it is not relevant to the current study. Consequently, we are focusing on the molecules that are attached to the membrane and excluding nonspecific binding using our method.

1. Tang, B., Chong, K., Masefski, W., & Evans, R. (2022). Quantitative Interpretation of Protein Diffusion Coefficients in Mixed Protiated-Deuterated Aqueous Solvents. The journal of physical chemistry. B, 126(31), 5887–5895. <https://doi.org/10.1021/acs.jpcc.2c03554>

4. On the other hand, the confocal FRAP in Figure 1I-J looks promising. Can we have FRAP data also on the receptor-neonG channel, please?

Response. The revisited version shows this data in figure S7.

5. The explanation of the mechanism hinges on Figure S6, which, as discussed below, raised some issues of consistency with other data presented in the manuscript. Crucially, it would be now important to conduct FRAP experiments on cells treated with hyaluronidase.

Response. This experiment is challenging because the density of PTH^{TMR} molecules after HAase treatment is too low to allow accurate measurements.

6. Figure S6A dissociation of the ligand: in native conditions, most ligand is dissociated within 2 minutes. However, there is recovery at 200 s (3 minutes) shown in Figure 1I. How do you interpret FRAP data and mobile fractions of a rapidly dissociating ligand?

Response. Figure S6A shows ligand dissociation from the plasma membrane with no expression of PTH1R. However, the FRAP experiment in Figure 1I quantifies lateral diffusion in cells expressing the receptor.

Reviewer #4 (Remarks to the Author):

Pacheco, Vilardaga, and colleagues present an interesting study in which they use single molecule TIRF imaging—with supporting biochemistry—to argue against the classic text-book model of GPCR activation. Instead of the ligand diffusing through solution to the receptor, the authors show that fluorophore-labeled PTH associates with the cell membrane and the encounter with PTH1R is through diffusion of the receptor to the relatively immobile ligand. Again, unlike the text-book model, the authors argue that the primary association of the receptor-ligand complex with B-arrestin is at the CCP (rather than association of the tripartite complex prior to CCP entry). The study is impressive (the inverted TIRF setup for imaging the top of the cell is particularly clever) and extensive controls have been added during previous rounds of review. Of particular note are the line of new control experiments depleting hyaluronic acid and showing reduced TMR-PTH association with the cell membrane (it would be nice if these data weren't buried in supplement, as they are an important control).

I think this is an excellent study and I would recommend it for publication after addressing my few relatively minor criticisms.

1) Figure 4. I am surprised that formation of the B-arrestin/PTH1R complex is fully blocked with TGX-221 treatment (Figure 4C). The implication from Figure 4 is that agonist-induced endocytosis of PTH1R would also be fully blocked by TGX-211? I thought the important control experiment (PTH1R endocytosis assays using TGX-211) might be present in this manuscript or White et al., 2020, but I couldn't find it. Can the authors address this concern? Ideally, with flow-cytometry based assays for agonist-induced PTH1R endocytosis with and without TGX-211. My concern is that—as the authors know—many non-Gq coupled receptors recruit B-arrestin (presumably without the PIP3 mechanism described here and in White et al. 2020). I would have assumed that those mechanisms (Gbeta/gamma with GRK2/3, C-edge, etc) could still come into play here even

if PIP3 levels aren't increased and that these mechanisms would still allow for PTH1R/B-arrestin association and subsequent endocytosis (albeit, at lower levels than with Gq/PIP3). If the authors find that agonist-induced endocytosis of PTH1R isn't fully blocked by TGX-211 treatment, the implication is that there are other—biologically relevant—PTH1R/B-arrestin interactions the authors aren't capturing in their single molecule assays, which would be important to note in the text.

Response. The reviewer is right. There is a β arrestin (and TGX211)-independent PTH1R internalization pathway. We addressed this point as follows: “An implication from the results is that the process of PTH1R internalization triggered by PTH could be blocked by TGX-211. We examined this hypothesis by using PTH1R fused to super-ecliptic pHluorin, a pH-sensitive variant of GFP. Our findings indicate that PTH-activated PTH1R internalized less and has a quicker recycling process in cells treated with TGX-221, or in cells lacking expression of β -arrestin isoforms 1 and 2 (Fig. 4E). Therefore, TGX-221 was not able to entirely block PTH1R internalization, which occurred via a β arr-independent internalization of PTH1R. (page 10/11 and Figure 4E).

Fig. PTH1R internalization. (A) schematic depicting the experiment. (B) Time courses of internalization and recycling of the PTH_{1R} tagged with super-ecliptic pHluorin (PTH^{SEP}) that exhibits fluorescence intensity reduction in acidic environments in response to PTH in HEK293 cells preincubated with DMSO (control, black) or 100 nM TGX-221 (pink) for 30 min before live cell imaging control and in β arr2-KO HEK293 cells, measured by time-lapse confocal microscopy. The baseline was established, and cells were perfused with a 100 nM PTH (horizontal bar) then washed out. Data are means \pm SD from $N = 2-3$ experiments and $n \geq 100$ cells per condition. (C) Expression of β -arrestins in parental and β arrestin-KO HEK-293 cells using an anti- β arr1/2 antibody.

2) Better comparison with Grimes et al., Cell, 2023. One of the two critical findings of this paper is that B-arrestin is recruited to the PM and CCP (largely) independently of receptor-ligand (RL) complex. A similar finding was recently shown (Grimes et al., 2023) for B2AR. Considering the broad similarity of these studies and techniques, the authors discussion of Grimes—relative to their own paper—is important. However, I found the discussion about Grimes quite confusing. Please re-write/clarify this section of the discussion to underscore the similarities between the studies (B-arrestin is a largely independent of the receptor in its movement to CCPs) and highlight the differences regarding B-arrestin (C-edge vs PIP3, etc). Also, I found it surprising that Eichel et al., 2016 and 2018 weren't cited considering these studies showed early evidence for B-arrestin acting independently of RL at the CCP.

Response. More discussion has been included in page 16, highlighting the similarities and differences between both works. Additionally, the revisited manuscript cites Eichel et al.

3) Explanation of PIP-resin experiments. I find part of the PIP resin experiments confusing (Figure 5B). Can the authors offer a rational/information for:

A) Why activation with PTH added before lysis (and before at 18 hour pulldown) causes a persistent change in the ability of B-arrestin to binding PIP3 resin (lanes 3 and 4 much higher than lane 1 or 2 as seen Figure 5B and 5C). Do the authors think B-arrestin is activated through a PTM, and that is the persistent activating factor? I would assume any cell biological changes would likely have been long lost during the incubation with resin (and even if it wasn't not, changes like increased cellular PIP3 should have acted as a competitive inhibitor to binding PIP3 resin).

Response: The reviewer's point about questioning the PTH-dependency of the β arr-PIP₃ interaction is valid. We have discussed this point as follows: "The enhanced capability of β -arrestin to attach to the PIP₃ resin when PTH is present suggests that the interaction between β -arrestins and PIP₃ is strengthened by the PTH-bound receptor or facilitated by post-translational modification of β -arrestins (such as phosphorylation) induced by PTH. (page 12)"

B) Why is the total amount of PTHR-HA (input blot) greater in the PM and C fractions when agonist is added (lanes 3 and 4) compared to no agonist (lanes 1 and 2)? My concern here is that the data suggest that adding agonist improves the ability of B-arrestin to bind PIP3 resin (as shown in quantification in 5C, grey bars vs red); however, if there is a different amount of PTH1R in the 10cm plates, direct comparison of agonist and no-agonist could be misleading;

Response. The cells in both plates were lysed when they reached the same level of confluency. We ensured that an equal number of cells were seeded in both plates, and these cells have been stably transfected with PTH1R-HA, ensuring consistent receptor expression. While it's unlikely that PTH stimulation for 5 minutes would increase the cell count, we can't rule out the possibility that PTH could enhance either the translation or trafficking of PTH1R to the plasma membrane. Furthermore, the increased levels of PTH1R in the cytosolic fraction (lane 3 in input blot) are likely due to PTH-induced internalized PTH1R. However, our crude membrane/cytosolic fraction doesn't allow us to distinguish endosomal membranes from the cytosolic fraction. As mentioned in point A, it is possible that PTH increases the ability of β -arrestin to bind to PIP3. We are confident that initial PTH1R levels are even among conditions.

C) I couldn't find information in the material and methods on where the PIP resin was purchased (I assume Echelon Biosciences?).

Response. Correct. PIP3 beads were from Echelon Bioscience (Catalog # P-B003A). We added this information to the material and methods section (page 23).

4) Regarding previous rounds of review: The idea that PTH associates with the cell, and then encounters PTH1R through lateral diffusion of the receptor, is surprising. The authors have added many controls from the previous rounds of review to strengthen their argument, including: 1) the choice of fluorophore (TMR and fluorescein) doesn't alter PTH foci; 2) addition of unlabeled PTH also causes slowing of PTH1R; 3) a mechanistic explanation with hyaluronic acid manipulation. I am generally satisfied and do not think additional controls are required.

Response. We agree.

As previously, we are grateful for reviewer #3's thoughtful comments. We have carefully considered and addressed his/her comments in our responses (highlighted in yellow).

Reviewer #3

Pacheco et al have provided a revised version of their manuscript, which addressed some, but not all of the concerns that were addressed in a previous revision. Some of the new experiments that appear to better support their claim, are unfortunately missing some important controls. Since the suggestion in previous rounds of review had been quite specific, I take the liberty of reiterating the specific experiments that would allow to allay any remaining concern:

1. In the 'flipped geometry' the top membrane becomes again a bottom membrane, i.e. it is sandwiched and pushed against the coverslip, so any concern related to the original bottom membrane, would also apply here, such as geometrical ligand confinement. I therefore suggest to remove the connotation that these experiments report on the cell apical membrane.

Response. We greatly appreciate your comment, which allows us to include some information in the methodology. Our approach involved adding the fluorescent ligand to the cells first, then removing it (wash off). We then inverted the coverslip to implement the TIRF configuration. This method ensures the ligand's attachment to the membrane, preventing any residual ligand from floating. The revisited manuscript now provides a more detailed account of our experiment procedure on page 5

2. A very specific experiment had been suggested, i.e. the use of a different labeled 'mock' peptide to address the issue of the potential non specific binding. Ideally a peptide other than PTH, lacking the sequence BXXXXXB should be imaged by single particle tracking .

Response. The conclusions put forward by the reviewer may not be directly relevant to PTH behavior. We are seeking further understanding of whether the experiment involves randomization of the PTH sequence or another peptide unrelated to PTHR. Our findings indicate that PTH₁₋₁₄, which does not contain a potential BXXXXXB motif, exhibits a weak attachment to the cell surface compared to PTH.

Response2: The very reasonable request to use a different peptide, different from PTH, was meant to allay a purely methodological concern, related to the possibility that there could be a substantial degree of sticking of the peptide to the coverslip in the tight space between the cell membrane and said coverslip. These concerns did not originate out of thin air, but due to the data originally presented in 2023 as the manuscript was first reviewed in Nature Chemical Biology. I report here Figure S3 from that submission:

Figure 1 From Figure S3 Nat Chem Bio submission

As can be seen, the ligand not only sticks a lot to the coverslip alone, but also when 'mock' (i.e. no PTHR) cells are present. This figure should now be compared to Figure 1B from the current submission. As it can be seen, the issue of the ligand (PTH^{TMR}) sticking to coverslip is far from trivial. This is still present, albeit in a reduced form in the information provided in

Supplementary Figure 4 of the current revision, below:

Figure 2 From Figure S4 current submission

The PTH^{TMR} does stick to the coverglass, and a diffusion-based measurements are invoked to differentiate the two situations. For a seminal paper claiming that ligand immobilization at the plasma membrane precedes ligand-receptor binding, it is clear that one would like to have an experimental strategy that dispels any concerns related to these unwanted effects. For this reason, I took the liberty of emphasizing the FRAP experiments below.

I have the feeling we are circling these issues since over a year, and I do not really see the reason why the experiments that were suggested already over a year ago could not be implemented in the several revisions we have seen since.

We fully agree with your observations regarding the sedimentation of PTH-TMR at the bottom of the coverslip. As the reviewer rightly points out, this is not a trivial matter, a reason why we have included the best possible control in Figure 1D and 1E using a state-of-the-art approach that clearly demonstrates the cell surface diffusion of PTH was independent of cell attachment or non-specific ligand binding to the glass coverslip. We measured PTH-TMR diffusion in the top part of the cell (not attached to the glass coverslip) using TIRF microscopy, where the coverslip was held in a screw-down piston device to allow the adjustment of the top cell surface against the coverslip of a glass-bottom dish (Fig. 1D). This assay ensures the ligand's attachment to the coverslip is prevented. We obtained comparable diffusion coefficient values of PTH-TMR on both the bottom and upper parts of the cell membrane, thus validating our results obtained from the bottom side of cells.

Additionally, we demonstrate a clear difference in diffusion between the PTH bound to the cell surface and the glass coverslip (Figs. 1C and S5C). Therefore, the non-specific binding to the coverslip is easily differentiated from the specific binding at the cell surface.

This control is described as follows in page 5:

“To verify that the cell surface diffusion of PTH was independent of cell attachment to the coverslip, we measured PTH^{TMR} diffusion in the top part of the cell (not attached to the glass coverslip). To this end, we employed a live-cell technique, initially developed to image apical membrane proteins of epithelial cells in combination with TIRF microscopy, which provides the best signal-to-noise ratio suitable for single-molecule imaging (12). To conduct this experiment, the cells were cultured on coverslips, and PTH^{TMR} was added and washed out to prevent leftover ligands from floating around. Then, the coverslip was held in a screw-down piston device to allow the adjustment of the top cell surface against the coverslip of a glass-bottom dish (Fig. 1D). We obtained comparable diffusion coefficient values of PTH^{TMR} on both the bottom and upper parts of the cell membrane, thus validating our results obtained from the bottom side of cells. (Fig. 1E).”

For me, the following points remain outstanding:

1. Is the entity of receptor PTH^{TMR} immobilization to fibronectin slides even in the absence of PTHR on the surface of the cells (mock). Notably statistical analysis is missing in Figure S4C, but the AUC results suggest that the difference between PTHR cells and Mock cells is rather limited. Moreover, In Figure 1C the diffusion of PTH^{TMR} is not statistically larger in mock cells than in PTHR transfected cells

We apologize for not showing the asterisks in Figure S4C. Using a multiple comparison test, we found no significant differences in the AUC for Mock and PTH1R-expressing cells. This is expected, given the low expression conditions of PTH1R, which are suitable for single-molecule tracking. In agreement with this result, there were no significant differences in diffusion for PTH^{TMR} in mock or PTH1R-expressing cells. We interpret these results as indicating that most of the single-molecule measurements of PTH^{TMR} correspond to the unbound state and represent the intrinsic diffusion of the ligand. This finding has significant implications for our understanding of ligand diffusion.

2. The same marginal differences are observed for the, overall more convincing, FRAP experiments.

Figure 3 from Figure 11 current submission

Figure 4 From Figure S7 of current submission

The recovery in Figure 1I of the manuscript tells us that PTH^{TMR} is pretty immobile at the plasma membrane (not necessarily basal), i.e. 20% of the ligand can move around. Figure S7A however tells us that, in absence of the ligand, just 30% of the PTHR is mobile, suggesting that most of the receptors are immobile to begin with.

We are thus facing a situation where the single molecule data must contend with a non-negligible fraction of immobilization events on the coverslips, whereas the FRAP data navigate an incremental change of immobilization of an already largely immobile population of receptors.

Moreover, the ligand devoid of the putative bridging motif PTH₁₋₁₄^{TMR} displays a diffusion coefficient (small, meaning largely immobile) comparable to that of the full ligand. So, albeit less of this ligands binds (according to the data in Figure S6), once bound it displays a similar diffusion rate than the PTH^{TMR}. How would the authors reconcile this with their hypothesis?

Figure 5 From Figure 2I of the current submission

In this context, it appears only reasonable to ask for the mobility behavior of a different peptide ligand both in mock as well as PTHR expressing cells.

If the authors feel that the PTH₁₋₁₄^{TMR} is indeed the right control (i.e. lacks the domain bridging interaction with the surface glycans, despite the data of Figure 2I), then please do the FRAP experiments using this ligand. If the recovery is of the order of 30%, or significantly larger than 20%, then it means that the immobilization mediated by the BXXXXXXB motif is not real. On the other hand, if the recovery is again 20%, then all is good and the hypothesis of the authors is supported.

We appreciate the reviewer's insightful interpretation of the FRAP results. The idea that many immobile receptors could act as reservoirs for ligands through direct interaction was not considered. However, it is important to emphasize that the FRAP results indicate a phenomenon different from the single-molecule (SM) data.

In FRAP experiments, the receptor is overexpressed to ensure there is enough ligand at the cell surface. On the other hand, for SM tracking, PTHR expression is kept low to achieve a low molecular density. In this context, FRAP of PTH^{TMR} showed the slow diffusion of the PTH-PTHR complex at the ensemble level. This was supported by the similar mobile fraction obtained for both the active receptor and PTH^{TMR} (Fig. 1J and S7B).

This led us to measure the diffusional properties of PTH in cells overexpressing the receptor. We anticipated that high PTHR expression would favor direct association with the ligand, bypassing pre-association with the plasma membrane and showing more movement than cells with lower expression. The results of that experiment shown in Fig. S8 confirm this hypothesis.

Fig. S8| Overexpression of PTHR increases the diffusion of PTH^{TMR}. (A) Diffusion coefficient of PTH^{TMR} in HEK293 cells with high and low expression of PTHR^{mNG}, and in cell lines with endogenous expression of PTHR (MC3T3 and RPTEC cells). Similar values were also obtained on the top membrane of HEK293 cells with low expression of PTHR. (B) Example of trajectories of PTH^{TMR} in a cell with high expression of PTHR^{mNG}. (C, D) Diffusion (C) and motion (D) time-lapses of PTH^{TMR} in cells overexpressing PTHR (black) and in mock cells (magenta). Individual trajectories were analyzed at intervals of 10 seconds. Ligands were added at T = 30 s (vertical dotted line). Data are the mean ± S.E.M.

Regarding the PTH1-14^{TMR} lacking the BXXXXXB motif, we emphasize that our data supports a model where hyaluronic acid is important for binding PTH ligands to the plasma membrane. However, the precise mechanism for confined molecules and, thus, their diffusion remains to be explored.

3. Use of diffusion to discriminate ligand bound/stuck to surface from ligand embedded in the membrane is not conclusive, since the ligand diffusing in the gap between the cell membrane and the coverslips (either bottom or top) could have an additional mode of motion.

Response. We understand that there may be some amount of ligand present in the aqueous phase. However, in our particular case, this amount is not significant because several studies have shown that diffusion in aqueous phases is >10 μm²/s (1), which is much faster than our detection speed. Therefore, it is not relevant to the current study. Consequently, we are focusing on the molecules that are attached to the membrane and excluding nonspecific binding using our method.

1. Tang, B., Chong, K., Masefski, W., & Evans, R. (2022). Quantitative Interpretation of Protein Diffusion Coefficients in Mixed Protiated-Deuteriated Aqueous Solvents. *The journal of physical chemistry. B*, 126(31), 5887–5895. <https://doi.org/10.1021/acs.jpccb.2c03554>

Response 2 This is not the case, since the PTH^{TMR} may be hopping from one site to the other, either on the cell or on the coverslip. Your detection speed in this case, is not particularly relevant.

We disagree with the reviewer's comment. In cases involving the unbinding and binding events of the peptide (hopping), the molecule must diffuse rapidly during the hopping process, similar to its behavior in the aqueous phase. As a result, it becomes undetectable due to our acquisition speed. For instance, successful single-molecule tracking of lipid binding domains (LBDs) labeled with fluorescent proteins has been reported (2). In this case, LBDs exhibit an affinity that ranges from nano- to micromolar, leading to transient binding to the plasma membrane (PM). However, hopping events are undetectable because the diffusion of the sensors, when not bound to the PM, exceeds the sampling frequency.

Ref 2. Pacheco J, Cassidy AC, Zewe JP, Wills RC, Hammond GRV. PI(4,5)P2 diffuses freely in the plasma membrane even within high-density effector protein complexes. J Cell Biol. 2023 Feb 6;222(2):e202204099. PMID: 36416724; PMCID: PMC9698391.

4. On the other hand, the confocal FRAP in Figure 1I-J looks promising. Can we have FRAP data also on the receptor-neonG channel, please?

Response. The revisited version shows this data in figure S7.

Response2 Receptor's diffusion in basal condition (Figure S7A,B) is limited in the first place, i.e. 30% recovery, which goes down to 20% when PTH is added. This value matches what is observed in Figure 1I for PTH-TMR

We agree; check the previous answer to point 2.

5. The explanation of the mechanism hinges on Figure S6, which, as discussed below, raised some issues of consistency with other data presented in the manuscript. Crucially, it would be now important to conduct FRAP experiments on cells treated with hyaluronidase.

Response. This experiment is challenging because the density of PTH^{TMR} molecules after HAase treatment is too low to allow accurate measurements.

Response2 According to the data in Figure S6A you should have at least 5% of your ligand still bound after HA treatment in untransfected cells, but my suggestion is to do this in transfected cells. In this instance, you should at least see the difference between the ligand pre-immobilised on the membrane, in turn immobilising the receptor (-HA), and the ligand just bound to the receptor (+HA).

This is not the case, given that overexpression of PTH₁R favors a direct association with the ligand, bypassing pre-association with the plasma membrane:

6. Figure S6A dissociation of the ligand: in native conditions, most ligand is dissociated within 2 minutes. However, there is recovery at 200 s (3 minutes) shown in Figure 1I. How do you interpret FRAP data and mobile fractions of a rapidly dissociating ligand?

Response. Figure S6A shows ligand dissociation from the plasma membrane with no expression of PTH₁R. However, the FRAP experiment in Figure 1I quantifies lateral diffusion in cells

We are grateful for the thoughtful comments provided by the reviewers. We have taken these comments into careful consideration and ensured that all concerns have been thoughtfully addressed.

Reviewer #3

Pacheco et al have provided a revised version of their manuscript, which addressed some, but not all of the concerns that were addressed in a previous revision. Some of the new experiments that appear to better support their claim, are unfortunately missing some important controls. Since the suggestion in previous rounds of review had been quite specific, I take the liberty of reiterating the specific experiments that would allow to allay any remaining concern:

1. In the 'flipped geometry' the top membrane becomes again a bottom membrane, i.e. it is sandwiched and pushed against the coverslip, so any concern related to the original bottom membrane, would also apply here, such as geometrical ligand confinement. I therefore suggest to remove the connotation that these experiments report on the cell apical membrane.

Response. We greatly appreciate your comment, which allows us to include some information in the methodology. Our approach involved adding the fluorescent ligand to the cells first, then removing it (wash off). We then inverted the coverslip to implement the TIRF configuration. This method ensures the ligand's attachment to the membrane, preventing any residual ligand from floating. The revisited manuscript now provides a more detailed account of our experiment procedure on page 5

2. A very specific experiment had been suggested, i.e. the use of a different labeled 'mock' peptide to address the issue of the potential non specific binding. Ideally a peptide other than PTH, lacking the sequence BXXXXXB should be imaged by single particle tracking .

Response. The conclusions put forward by the reviewer may not be directly relevant to PTH behavior. We are seeking further understanding of whether the experiment involves randomization of the PTH sequence or another peptide unrelated to PTHR. Our findings indicate that PTH₁₋₁₄, which does not contain a potential BXXXXXB motif, exhibits a weak attachment to the cell surface compared to PTH.

Response2: The very reasonable request to use a different peptide, different from PTH, was meant to allay a purely methodological concern, related to the possibility that there could be a substantial degree of sticking of the peptide to the coverslip in the tight space between the cell membrane and said coverslip. These concerns did not originate out of thin air, but due to the data originally presented in 2023 as the manuscript was first reviewed in Nature Chemical Biology. I report here Figure S3 from that submission:

Figure 1 From Figure S3 Nat Chem Bio submission

As can be seen, the ligand not only sticks a lot to the coverslip alone, but also when 'mock' (i.e. no PTHR) cells are present. This figure should now be compared to Figure 1B from the current submission. As it can be seen, the issue of the ligand (PTH^{TMR}) sticking to coverslip is far from trivial. This is still present, albeit in a reduced form in the information provided in

Supplementary Figure 4 of the current revision, below:

Figure 2 From Figure S4 current submission

The PTH^{TMR} does stick to the coverglass, and a diffusion-based measurements are invoked to differentiate the two situations. For a seminal paper claiming that ligand immobilization at the plasma membrane precedes ligand-receptor binding, it is clear that one would like to have an experimental strategy that dispels any concerns related to these unwanted effects. For this reason, I took the liberty of emphasizing the FRAP experiments below.

I have the feeling we are circling these issues since over a year, and I do not really see the reason why the experiments that were suggested already over a year ago could not be implemented in the several revisions we have seen since.

For me, the following points remain outstanding:

1. Is the entity of receptor PTH^{TMR} immobilization to fibronectin slides even in the absence of PTHR on the surface of the cells (mock). Notably statistical analysis is missing in Figure S4C, but the AUC results suggest that the difference between PTHR cells and Mock cells is rather limited. Moreover, In Figure 1C the diffusion of PTH^{TMR} is not statistically larger in mock cells than in PTHR transfected cells
2. The same marginal differences are observed for the, overall more convincing, FRAP experiments.

Figure 3 from Figure 11 current submission

Figure 4 From Figure S7 of current submission

The recovery in Figure 11 of the manuscript tells us that PTH^{TMR} is pretty immobile at the plasma membrane (not necessarily basal), i.e. 20% of the ligand can move around. Figure S7A however tells us that, in absence of the ligand, just 30% of the PTHR is mobile, suggesting that most of the receptors are immobile to begin with.

We are thus facing a situation where the single molecule data must contend with a non-negligible fraction of immobilization events on the coverslips, whereas the FRAP data navigate an incremental change of immobilization of an already largely immobile population of receptors.

Moreover, the ligand devoid of the putative bridging motif PTH₁₋₁₄^{TMR} displays a diffusion coefficient (small, meaning largely immobile) comparable to that of the full ligand. So, albeit less of this ligand binds (according to the data in Figure S6), once bound it displays a similar diffusion rate than the PTH^{TMR}. How would the authors reconcile this with their hypothesis?

Figure 5 From Figure 21 of the current submission

In this context, it appears only reasonable to ask for the mobility behavior of a different peptide ligand both in mock as well as PTHR expressing cells.

If the authors feel that the PTH₁₋₁₄^{TMR} is indeed the right control (i.e. lacks the domain bridging interaction with the surface glycans, despite the data of Figure 21), then please do the FRAP experiments using this ligand. If the recovery is of the order of 30%, or significantly larger than 20%, then it means that the immobilization mediated by the BXXXXXXB motif is not real. On the other hand, if the recovery is again 20%, then all is good and the hypothesis of the authors is supported.

3. Use of diffusion to discriminate ligand bound/stuck to surface from ligand embedded in the membrane is not conclusive, since the ligand diffusing in the gap between the cell membrane and the coverslips (either bottom or top) could have an additional mode of motion.

Response. We understand that there may be some amount of ligand present in the aqueous phase. However, in our particular case, this amount is not significant because several studies have shown that diffusion in aqueous phases is $>10 \mu\text{m}^2/\text{s}$ (1), which is much faster than our detection speed. Therefore, it is not relevant to the current study. Consequently, we are focusing on the molecules that are attached to the membrane and excluding nonspecific binding using our method.

1. Tang, B., Chong, K., Masefski, W., & Evans, R. (2022). Quantitative Interpretation of Protein Diffusion Coefficients in Mixed Protiated-Deuteriated Aqueous Solvents. *The journal of physical chemistry. B*, 126(31), 5887–5895. <https://doi.org/10.1021/acs.jpcc.2c03554>

Response 2 This is not the case, since the PTH^{TMR} may be hopping from one site to the other, either on the cell or on the coverslip. Your detection speed in this case, is not particularly relevant.

4. On the other hand, the confocal FRAP in Figure 1I-J looks promising. Can we have FRAP data also on the receptor-neonG channel, please?

Response. The revisited version shows this data in figure S7.

Response2 Receptor's diffusion in basal condition (Figure S7A,B) is limited in the first place, i.e. 30% recovery, which goes down to 20% when PTH is added. This value matches what is observed in Figure 1I for PTH-TMR

5. The explanation of the mechanism hinges on Figure S6, which, as discussed below, raised some issues of consistency with other data presented in the manuscript. Crucially, it would be now important to conduct FRAP experiments on cells treated with hyaluronidase.

Response. This experiment is challenging because the density of PTH^{TMR} molecules after HAase treatment is too low to allow accurate measurements.

Response2 According to the data in Figure S6A you should have at least 5% of your ligand still bound after HA treatment in untransfected cells, but my suggestion is to do this in transfected cells. In this instance, you should at least see the difference between the ligand pre-immobilised on the membrane, in turn immobilising the receptor (-HA), and the ligand just bound to the receptor (+HA).

6. Figure S6A dissociation of the ligand: in native conditions, most ligand is dissociated within 2 minutes. However, there is recovery at 200 s (3 minutes) shown in Figure 1I. How do you interpret FRAP data and mobile fractions of a rapidly dissociating ligand?

Response. Figure S6A shows ligand dissociation from the plasma membrane with no expression of PTH1R. However, the FRAP experiment in Figure 1I quantifies lateral diffusion in cells expressing the receptor.

Reviewer #4 (Remarks to the Author):

Pacheco, Vilardaga, and colleagues present an interesting study in which they use single

molecule TIRF imaging—with supporting biochemistry—to argue against the classic text-book model of GPCR activation. Instead of the ligand diffusing through solution to the receptor, the authors show that fluorophore-labeled PTH associates with the cell membrane and the encounter with PTH1R is through diffusion of the receptor to the relatively immobile ligand. Again, unlike the text-book model, the authors argue that the primary association of the receptor-ligand complex with B-arrestin is at the CCP (rather than association of the tripartite complex prior to CCP entry). The study is impressive (the inverted TIRF setup for imaging the top of the cell is particularly clever) and extensive controls have been added during previous rounds of review. Of particular note are the line of new control experiments depleting hyaluronic acid and showing reduced TMR-PTH association with the cell membrane (it would be nice if these data weren't buried in supplement, as they are an important control).

I think this is an excellent study and I would recommend it for publication after addressing my few relatively minor criticisms.

1) Figure 4. I am surprised that formation of the B-arrestin/PTH1R complex is fully blocked with TGX-221 treatment (Figure 4C). The implication from Figure 4 is that agonist-induced endocytosis of PTH1R would also be fully blocked by TGX-221? I thought the important control experiment (PTH1R endocytosis assays using TGX-211) might be present in this manuscript or White et al., 2020, but I couldn't find it. Can the authors address this concern? Ideally, with flow-cytometry based assays for agonist-induced PTH1R endocytosis with and without TGX-211. My concern is that—as the authors know—many non-Gq coupled receptors recruit B-arrestin (presumably without the PIP3 mechanism described here and in White et al. 2020). I would have assumed that those mechanisms (Gbeta/gamma with GRK2/3, C-edge, etc) could still come into play here even if PIP3 levels aren't increased and that these mechanisms would still allow for PTH1R/B-arrestin association and subsequent endocytosis (albeit, at lower levels than with Gq/PIP3). If the authors find that agonist-induced endocytosis of PTH1R isn't fully blocked by TGX-211 treatment, the implication is that there are other—biologically relevant—PTH1R/B-arrestin interactions the authors aren't capturing in their single molecule assays, which would be important to note in the text.

Response. The reviewer is right. There is a β arrestin (and TGX211)-independent PTH1R internalization pathway. We addressed this point as follows: “An implication from the results is that the process of PTH1R internalization triggered by PTH could be blocked by TGX-211. We examined this hypothesis by using PTH1R fused to super-ecliptic pHluorin, a pH-sensitive variant of GFP. Our findings indicate that PTH-activated PTH1R internalized less and has a quicker

recycling process in cells treated with TGX-221, or in cells lacking expression of β -arrestin isoforms 1 and 2 (Fig. 4E). Therefore, TGX-221 was not able to entirely block PTH1R internalization, which occurred via a β arr-independent internalization of PTH1R. (page 10/11 and Figure 4E).

2) Better comparison with Grimes et al., Cell, 2023. One of the two critical findings of this paper is that B-arrestin is recruited to the PM and CCP (largely) independently of receptor-ligand (RL) complex. A similar finding was recently shown (Grimes et al.,2023) for B2AR. Considering the broad similarity of these studies and techniques, the authors discussion of Grimes—relative to their own paper—is important. However, I found the discussion about Grimes quite confusing. Please re-write/clarify this section of the discussion to underscore the similarities between the studies (B-arrestin is a largely independent of the receptor in its movement to CCPs) and highlight the differences regarding B-arrestin (C-edge vs PIP3, etc). Also, I found it surprising that Eichel et al., 2016 and 2018 weren't cited considering these studies showed early evidence for B-arrestin acting independently of RL at the CCP.

Response. More discussion has been included in page 16, highlighting the similarities and differences between both works. Additionally, the revisited manuscript cites Eichel et al.

3) Explanation of PIP-resin experiments. I find part of the PIP resin experiments confusing (Figure 5B). Can the authors offer a rational/information for:

A) Why activation with PTH added before lysis (and before at 18 hour pulldown) causes a persistent change in the ability of B-arrestin to binding PIP3 resin (lanes 3 and 4 much higher than lane 1 or 2 as seen Figure 5B and 5C). Do the authors think B-arrestin is activated through a PTM, and that is the persistent activating factor? I would assume any cell biological changes would likely have been long lost during the incubation with resin (and even if it wasn't not, changes like increased cellular PIP3 should have acted as a competitive inhibitor to binding PIP3 resin).

Response: The reviewer's point about questioning the PTH-dependency of the β arr-PIP₃ interaction is valid. We have discussed this point as follows: “The enhanced capability of β -arrestin to attach to the PIP₃ resin when PTH is present suggests that the interaction between β -arrestins and PIP₃ is strengthened by the PTH-bound receptor or facilitated by post-translational modification of β -arrestins (such as phosphorylation) induced by PTH. (page 12)”

B) Why is the total amount of PTHR-HA (input blot) greater in the PM and C fractions when agonist is added (lanes 3 and 4) compared to no agonist (lanes 1 and 2)? My concern here is that the data suggest that adding agonist improves the ability of B-arrestin to bind PIP3 resin (as shown in quantification in 5C, grey bars vs red); however, if there is a different amount of PTH1R in the 10cm plates, direct comparison of agonist and no-agonist could be misleading;

Response. The cells in both plates were lysed when they reached the same level of confluency. We ensured that an equal number of cells were seeded in both plates, and these cells have been stably transfected with PTH1R-HA, ensuring consistent receptor expression. While it's unlikely that PTH stimulation for 5 minutes would increase the cell count, we can't rule out the possibility that PTH could enhance either the translation or trafficking of PTH1R to the plasma membrane. Furthermore, the increased levels of PTH1R in the cytosolic fraction (lane 3 in input blot) are likely due to PTH-induced internalized PTH1R. However, our crude membrane/cytosolic fraction doesn't allow us to distinguish endosomal membranes from the cytosolic fraction. As mentioned in point A, it is possible that PTH increases the ability of β -arrestin to bind to PIP3. We are confident that initial PTH1R levels are even among conditions.

C) I couldn't find information in the material and methods on where the PIP resin was purchased (I assume Echelon Biosciences?).

Response. Correct. PIP3 beads were from Echelon Bioscience (Catalog # P-B003A). We added this information to the material and methods section (page 23).

4) Regarding previous rounds of review: The idea that PTH associates with the cell, and then encounters PTH1R through lateral diffusion of the receptor, is surprising. The authors have added many controls from the previous rounds of review to strengthen their argument, including: 1) the choice of fluorophore (TMR and fluorescein) doesn't alter PTH foci; 2) addition of unlabeled PTH also causes slowing of PTH1R; 3) a mechanistic explanation with hyaluronic acid manipulation. I am generally satisfied and do not think additional controls are required.

Response. We agree.